# Mixed Channel Dependency Diffusion Model with Retrieval Guidance for Time Series Forecasting

## Abstract

Recent advancements in deep learning techniques have improved the performance of time series forecasting, especially with state-of-the-art generative models. Despite making progress in capturing conditional time-series patterns with uncertainty, existing time series generative models face reliability and computational challenges in long-term forecasting, especially when the number of variate is large. Moreover, the maximum likelihood objective of generative modeling is prone to underestimation for low-density region of the data manifold, therefore leading to sub-optimal conditional sampling quality. In this paper, we propose a mixed channel dependency diffusion model with retrieval guidance (MiDDiR) to address these challenges. In MiDDiR, we employs a novel mixed channel dependency method on time series diffusion model, encoding historical time series in a channel-dependent manner to obtain informative historical representation while denoising in a channel-independent manner to decrease modeling complexity. During inference, we retrieve similar history occurrence for explicitly tilting the score estimation as retrieval guidance to enhance forecasting quality. Extensive experiments demonstrate the effectiveness of MiDDiR, outperforming baselines in a variety of real-world time series datasets.

## 1 Introduction

Time series forecasting is a fundamental task driving a variety of crucial real-world areas, including but not limited to energy Alvarez et al. (2010), meteorology Gneiting & Raftery (2005), traffic Lana et al. (2018), healthcare Morid et al. (2023) and finance Jacob & Smith (1972). Recent advancements in deep learning have made significant progress in building high-accuracy time series forecasting models, especially through generative modeling such as generative transformers Vaswani et al. (2017) and denoising diffusion probabilistic models (DDPMs) Ho et al. (2020). By considering the time series forecasting problem as conditional time series generation, generative models extend the forecasting scope from pointing forecasting settings to probabilistic forecasting setting, which involve uncertainty estimation that is meaningful in real world applications.

Despite the progress in applying generative models for generating realistic forecasts, there remains significant challenges in long-term time series forecasting where the forecasting horizon may cover tens or hundreds of time steps. Real-world time series usually involves multiple correlated variables, which is also referred to as multivariate time series. As the forecasting horizon grows, the number of individual observation points to predict in a multivariate time series samples increases by a factor of the number of variables, which imposes a high modeling complexity. For auto-regressive generation methods Rasul et al. (2021a); Fan et al. (2024), forecasting error tends to accumulate along time steps, making the long-term forecast less reliable. While non-autoregressive generation methods Tashiro et al. (2021); Shen & Kwok (2023) tempts to avoid error accumulation, generating long-term multivariate time series as one sample is similar to synthesizing high-resolution images, which faces computational challenges in accurately approximating high-dimensional probability distribution.

Moreover, given the non-stationary nature of real-world time series, a longer forecasting horizon implies more diversified patterns within each time series samples, making it more difficult for a

generative model to capture the underlying conditional distribution. While existing work has proposed reversible instance normalization (RevIN) Kim et al. (2022) and non-stationary transformer Liu et al. (2022); Ye et al. (2025); Li et al. (2024) for addressing non-stationarity, another challenge lies in capturing the low-density regions of the data manifold. This challenge stems from the inherent bias within data, where some rare while repeatable scenarios cause temporary distribution shift. Due to the maximum likelihood nature of typical generative model training, the low-density regions are prone to underfitting, resulting in sub-optimal performance of these models.

To solve the above challenges, we propose a mixed channel dependency diffusion model with retrieval guidance (MiDDiR) for time series forecasting. To summarize, the main contributions of this paper are listed as follows:

- MiDDiR employs a novel mixed channel dependency method on time series diffusion model, encoding historical time series in a channel-dependent manner to obtain informative representation while denoising in a channel independent manner to decrease modeling complexity.

- MiDDiR introduces retrieval guidance, retrieving similar history occurrence for explicitly tilting the sampling score to enhance forecasting quality. To the best of our knowledge, this work is the first attempt to analytically guide generation with retrieval.

- Extensive experiments show that MiDDiR outperforms existing baselines across both probabilistic forecasting and point forecasting metrics.

## 2 RELATED WORK

### 2.1 LONG-TERM MULTIVARIATE TIME SERIES FORECASTING

Recent progress in deep learning has improved long-term multivariate time series forecasting models with diverse architectures, including recurrent neural networks (RNNs) Salinas et al. (2020); Lai et al. (2018), temporal convolution networks (TCNs) Sen et al. (2019), multi-layer percep-trons (MLPs) Chen et al. (2023); Wang et al. (2024), and Transformers Nie et al. (2023); Zhou et al. (2022a); Wu et al. (2021); Zhou et al. (2021); Liu et al. (2024). Besides employing different basic architectures, these works explore utilizing specific properties of time series for improving forecasting performance, such as trend-seasonal decomposition Wang et al. (2024); Zhou et al. (2022a); Wu et al. (2021; 2023), multi-granularity Hou et al. (2022). Some have also leveraged modeling techniques like Mixture-of-experts (MoE) Zhou et al. (2022a;b) for better utilizing time series components, channel-independent modeling Nie et al. (2023) for higher model efficiency and data stationarization Kim et al. (2022); Liu et al. (2022). In contrast to these conventional modeling technique that focus on providing point forecast, MiDDiRprovides probabilistic forecasting for long-term multivariate time series.

### 2.2 GENERATIVE TIME SERIES FORECASTING

Probabilistic time series forecasting with generative models that directly capture the conditional time series distribution have garnered great attention in the literature. Earlier works employ copula Salinas et al. (2019) or normalizing flows de Bézenac et al. (2020); Rasul et al. (2021b) to approximate the multivariate distribution. Recently, diffusion models Ho et al. (2020) have proved to be strong in probabilistic modeling and are quickly adapted to time series area. Forecasting methods with diffusion model can be further categorized into auto-regressive methods Rasul et al. (2021a); Fan et al. (2024) and non-autoregressive methods Tashiro et al. (2021); Shen & Kwok (2023); Shen et al. (2024); Ye et al. (2025); Kollovieh et al. (2023); Li et al. (2024). Auto-regressive methods generate the cross-sectional multivariate predictions at each time, while non-autoregressive methods generate values of all variables for every time slots in the forecasting horizon as one sample. Compared with previous diffusion-based models, MiDDiRfirst employs mixed channel dependency strategy for decreasing distribution modeling complexity and introduces retrieval guidance technique to enhance conditional sampling quality.

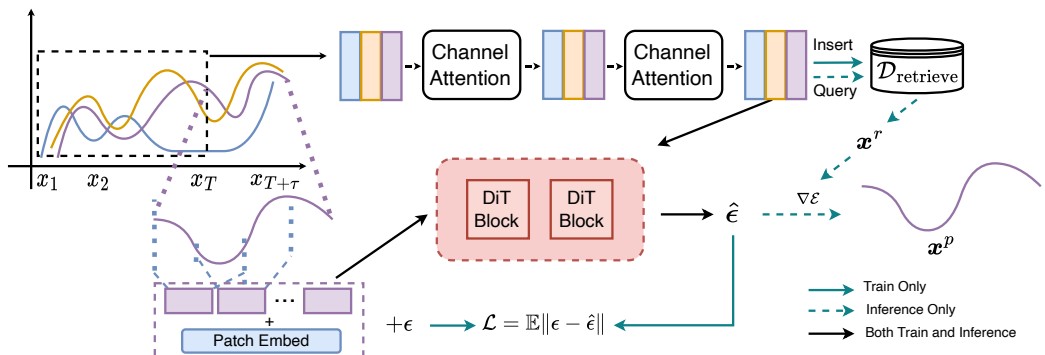

Figure 1: Overview of MiDDiR model.

# 3 METHODOLOGY

## 3.1 NON-AUTOREGRESSIVE HISTORY-CONDITIONED GENERATIVE TIME SERIES FORECASTING

In the context of probabilistic time series forecasting, given multivariate historical time series $\boldsymbol{X}^{\mathrm{o}} \in \mathbb{R}^{T \times C} = \{\boldsymbol{x}_1, ..., \boldsymbol{x}_T\}$ with $\boldsymbol{x}_t \in \mathbb{R}^C$, the subsequent future $\tau$-interval time series $\boldsymbol{X}^{\mathrm{p}} = \{\boldsymbol{x}_{T+1}, ..., \boldsymbol{x}_{T+\tau}\}$ follows a probabilistic distribution $q(\boldsymbol{X}^{\mathrm{p}}|\boldsymbol{X}^{\mathrm{o}})$. Instead of predicting next time step iteratively given previous time steps, a non-autoregressive forecasting model predicts the future time series as a whole. Therefore, it is convenient leveraging diffusion models to approximate the conditional distribution by optimizing:

$$\min D\left(q(\boldsymbol{X}^{\mathrm{p}}|\boldsymbol{X}^{\mathrm{o}})\|p_\theta(\boldsymbol{X}^{\mathrm{p}}|\boldsymbol{X}^{\mathrm{o}})\right), \tag{1}$$

where $\theta$ represents learnable model parameters of a neural network.

Assuming $N$ diffusion steps, the forward process of common diffusion model adds Gaussian noise on the future series to be predicted:

$$q(\boldsymbol{X}_1^{\mathrm{p}}, ..., \boldsymbol{X}_N^{\mathrm{p}} \mid \boldsymbol{X}_0^{\mathrm{p}}) \coloneqq \prod_{n=1}^{N} q(\boldsymbol{X}_n^{\mathrm{p}}|\boldsymbol{X}_{n-1}^{\mathrm{p}}), \tag{2}$$

where $q(\boldsymbol{X}_n^{\mathrm{p}}|\boldsymbol{X}_{n-1}^{\mathrm{p}}) \coloneqq \mathcal{N}(\boldsymbol{X}_n^{\mathrm{p}}; \sqrt{1-\beta_n}\boldsymbol{X}_{n-1}^{\mathrm{p}}, \beta_n\mathbf{I})$ and $\{\beta_1, ..., \beta_N\}$ is the noise variance schedule. The reverse process can be defined as the $\theta$ parameterized conditional estimation of posterior:

$$p_\theta(\boldsymbol{X}_{0:N}^{\mathrm{p}}|\boldsymbol{X}^{\mathrm{o}}) = p(\boldsymbol{X}_N^{\mathrm{p}}) \prod_{n=1}^{N} p_\theta(\boldsymbol{X}_{n-1}^{\mathrm{p}})|\boldsymbol{X}_t^{\mathrm{p}}, \boldsymbol{X}^{\mathrm{o}}), \tag{3}$$

where $p(\boldsymbol{X}_N^{\mathrm{p}}) = \mathcal{N}(\boldsymbol{X}_N^{\mathrm{p}}; \mathbf{0}, \boldsymbol{I})$ and the model can be reparameterized to predict conditional noise:

$$p_\theta(\boldsymbol{X}_{n-1}^{\mathrm{p}}|\boldsymbol{X}_n^{\mathrm{p}}, \boldsymbol{X}^{\mathrm{o}}) = \mathcal{N}(\boldsymbol{X}_{n-1}^{\mathrm{p}}; \frac{1}{\sqrt{1-\beta_t}}\left(\boldsymbol{X}_n^{\mathrm{p}} - \frac{\beta_n}{\sqrt{1-\bar{\alpha}_n}}\boldsymbol{\epsilon}_\theta\left(\boldsymbol{X}_{n-1}^{\mathrm{p}}, t, \boldsymbol{X}^{\mathrm{o}}\right)\right), \boldsymbol{\sigma}_\theta), \tag{4}$$

where $\bar{\alpha}_n = \prod_{i=1}^{n}(1-\beta_i)$.

## 3.2 MIXED CHANNEL DEPENDENCY DIFFUSION MODEL FOR TIME SERIES

Modeling the joint distribution of all observation points in long-term time series while capturing inter-channel relationship among multiple variates is similar to modeling high-resolution figures, which suffers from striking balance between generation quality and computation efficiency. Moreover, the inherent non-stationary and evolving dynamics of real-world time series make it more difficult for the model to capture generalizable patterns.

Although channel-independent modeling strategy has been proved to be capable for effectively reducing costs for capturing inter-channel dependency and is widely adopted by recent time series

forecasting literature, some argue that it is still beneficial utilizing signals embedded within inter-channel dependencies.

In this paper, we propose a mixed channel dependency strategy for time series diffusion forecasting model. Figure 1 shows the general framework of our model. Specifically, we propose to encode history time series with channel-dependent encoding module, then separate the encoded time-variate latent matrix channel-wise to treat each vector as the exclusive conditioning signal for denoising channel-independently.

### 3.2.1 CHANNEL-DEPENDENT ENCODING

For encoding the historical time series context $\boldsymbol{X}^{\text{o}}$ into latent vectors considering channel dependency while keeping light computational overhead, we employ a simple-yet-effective architecture using fully connected layers for encoding sequential feature, and attention blocks for mixing channel representations. For input $\boldsymbol{X}^{\text{o}}$, linear projection layers are applied on the time dimension, followed by an attention mechanism in each block. The outputs are then mapped into latent vector by another final linear projection layer. Let $L$ be the number of encoder blocks and $\boldsymbol{W}_I \in \mathbb{R}^{T \times D}$ be the weights of input projection $\boldsymbol{e}^0 \in \mathbb{R}^{C \times D} = \boldsymbol{X}^{\text{o}} \boldsymbol{W}_I$, where $D$ is the dimension of hidden layer, the encoding process can be formalized as:

$$\boldsymbol{z}^l \in \mathbb{R}^{D \times D} = \text{GeLU}(\boldsymbol{e}^{l-1}\boldsymbol{W}^l + \boldsymbol{b}), \quad \boldsymbol{e}^l \in \mathbb{R}^{D \times D} = \text{softmax}(\frac{\boldsymbol{z}^l \boldsymbol{W}_Q^l (\boldsymbol{z}^l \boldsymbol{W}_K^l)^\top}{\sqrt{D}})\boldsymbol{z}\boldsymbol{W}_V^l + \boldsymbol{e}^{l-1}$$
(5)

where $l \in \{1, ..., L\}$, $\boldsymbol{b}^l \in \mathbb{R}^D$ is the bias vector, $\boldsymbol{W}^l, \boldsymbol{W}_K^l, \boldsymbol{W}_Q^l, \boldsymbol{W}_V^l \in \mathbb{R}^{D \times D}$ are the projection weights of time, key, query, and value respectively. Finally, the output of last block is projected by a final weight matrix $\boldsymbol{W}_O \in \mathbb{R}^{D \times H}$, obtaining $\boldsymbol{e} \in \mathbb{R}^{C \times H} = \boldsymbol{e}^L \boldsymbol{W}_O$ where $H$ is the dimension of context embeddings. Let $\phi(\cdot)$ denotes the full encoder, we use a subscript $c \in \{1, ..., C\}$ to notate a the channel-specific embedding:

$$\boldsymbol{e} = \phi(\boldsymbol{X}^{\text{o}}) = \{\phi(\boldsymbol{X}^{\text{o}})_1, ..., \phi(\boldsymbol{X}^{\text{o}})_c, ..., \phi(\boldsymbol{X}^{\text{o}})_C\},$$
(6)

where $C$ is the number of channels. Hence, the historical time series context is encoded into channel-wise separated latent vectors.

### 3.2.2 CHANNEL-INDEPENDENT DENOISING

After encoding the historical time series context into channel-wise separated latent vectors, each of them are treated as the conditional signal for denoising a random sequence from prior distribution into the forecast for the corresponding channel. To capture sequential property for each single channel, we adopt a Diffusion Transformer (DiT)-like architecture with time step patching as our denoising network backbone.

Specifically, each input sequence is first divided into patches with patch size $p$. For better consistency, each patch is overlapped with coherent patches by half of the patch size $p/2$. Then the patches are leveraged as tokens for multi-head self-attention computation within each block. We employ learnable positional encoding with sinusoidal initialization, and use a separate learnable time step embedding model for embedding diffusion step as in common practice. The conditional signal from $\phi(\boldsymbol{X}^{\text{o}})_c$ is injected into the transformer blocks through zero-initialized adaptive layer normalization.

After all transformer blocks, the outputs is unpatched into the same shape of the input as the noise prediction. Omitting channel subscript for simplicity, the noise prediction is expressed as:

$$\hat{\boldsymbol{\epsilon}} = \boldsymbol{\epsilon}_\theta(\sqrt{\bar{\alpha}_n}\boldsymbol{x}_0^{\text{p}} + \sqrt{1 - \bar{\alpha}_n}\boldsymbol{\epsilon}, n, \phi(\boldsymbol{X}^{\text{o}})).$$
(7)

After predicting for all the channels, the noise vectors are concatenated along the channel dimension to construct the multivariate noise prediction.

### 3.3 END-TO-END TRAINING

The channel-dependent encoding module and the channel independent denoising module are trained in an end-to-end manner. The training objective is to minimize the distribution divergence between true and $\theta$-parameterized history-conditioned time series distribution as stated in eq. (1). Following

common practice in diffusion model training technique, we employ the $\epsilon$-parameterized loss function to maximize the evidence lower bound of the dataset log-likelihood for model training:

$$\max_{\theta} \log p_{\theta}(\boldsymbol{X}_0^{\mathrm{p}}|\boldsymbol{X}^{\mathrm{o}}) \implies \min_{\theta,\phi} \mathcal{L} = \mathbb{E}_{\boldsymbol{x}_0^{\mathrm{p}},n,\boldsymbol{X}^{\mathrm{o}},\boldsymbol{\epsilon}} \left[ \| \boldsymbol{\epsilon} - \boldsymbol{\epsilon}_{\theta}(\sqrt{\bar{\alpha}_n}\boldsymbol{x}_0^{\mathrm{p}} + \sqrt{1-\bar{\alpha}_n}\boldsymbol{\epsilon}, n, \phi(\boldsymbol{X}^{\mathrm{o}})) \|^2 \right], \tag{8}$$

where $\boldsymbol{\epsilon} \sim \mathcal{N}(\boldsymbol{0}, \boldsymbol{I})$, and $n \sim \mathrm{Uniform}(1, N)$.

Additionally, we apply reversible instance normalization (RevIN) technique to reduce non-stationarity. To be more specific, instance normalization statistics are calculated with the historical time series context $\boldsymbol{X}^{\mathrm{o}}$. However, in contrast to other models that apply reverse normalization on model output, we do not reverse the output of training phase since we are using $\epsilon$-prediction parameterization. Instead, we only reverse the final generation output of sampling phase. The pseudocode for training can be found at Algorithm 2.

### 3.4 IMPROVED SAMPLING WITH RETRIEVAL GUIDANCE

Real-world time series forecasting problem stems for fitting highly-complicated conditional distribution, where complicity lies in non-stationary dynamics and imbalanced data distribution. Although RevIN and the mixed dependency strategy help mitigate impact from non-stationary dynamics and are efficient for modeling long-term multivariate time series distribution, the maximum likelihood nature of diffusion model training may inherently underestimate the low-density regions of data manifold. When it comes to some underrepresented patterns that may occur rarely but repeatedly, the underestimation on these cases hurts the performance of the model.

We propose to apply retrieval guidance on diffusion sampling for mitigating this issue. To be more specific, we leverage the training set as source database to retrieve possible historic outcome given each test set context for referencing the denoising process. As a result, the performance on low-density region can be enhanced directly. The retrieval guidance process can further break down into two steps, namely mixed-dependency similarity retrieval and retrieval guidance score estimation.

#### 3.4.1 MIXED-DEPENDENCY SIMILARITY RETRIEVAL

Since the sampling process is channel independent, we retrieve guidance target in a channel-wise manner. For the retrieval key, we directly utilize the latent vectors generated by the channel-dependent encoder to maintain channel-dependent property of the embedded vectors, which is coherent with the mixed channel dependency strategy. Let $M$ be the number of individual multivariate training samples in training set $\mathcal{D}_{\mathrm{train}} = \{(\boldsymbol{X}_1^{\mathrm{o}}, \boldsymbol{X}_1^{\mathrm{p}}), ..., (\boldsymbol{X}_M^{\mathrm{o}}, \boldsymbol{X}_M^{\mathrm{p}})\}_{m=1,...,M}$, $C$ be the number of channels, the retrieval database $\mathcal{D}_{\mathrm{retrieval}}$ can be constructed following:

$$\{\boldsymbol{e}_{m\times 1}, ..., \boldsymbol{e}_{m\times C}\} = \phi(\boldsymbol{X}_m^{\mathrm{o}}), \quad \{\boldsymbol{x}_{m\times 1}^{\mathrm{p}}, ..., \boldsymbol{x}_{m\times C}^{\mathrm{p}}\} = \mathrm{Split}(\boldsymbol{X}_m^{\mathrm{p}}), \tag{9}$$

$$\mathcal{D}_{\mathrm{retrieval}} = \{(\boldsymbol{e}_1, \boldsymbol{x}_1^{\mathrm{p}}), ..., (\boldsymbol{e}_{M\times C}, \boldsymbol{x}_{M\times C}^{\mathrm{p}})\}. \tag{10}$$

When sampling, for each test set sample context $\boldsymbol{X}^{\mathrm{o}} \in \mathcal{D}_{\mathrm{test}}$, we first encode it into channel-wise latent vector $\boldsymbol{e}_c$, then retrieve top-$K$ closest latent vectors in $\mathcal{D}_{\mathrm{retrieval}}$ by cosine similarity, and finally calculate the average of the $K$ retrieved target time series weighted by similarity value:

$$i_1, ..., i_K = \arg \mathop{\mathrm{Top\text{-}K}}_{(\boldsymbol{e}_i, \boldsymbol{x}_i^{\mathrm{p}}) \in \mathcal{D}_{\mathrm{test}}} s_i, \quad \boldsymbol{x}_c^{\mathrm{r}} = \frac{\sum_{k=1}^K s_{i_k} \cdot \boldsymbol{x}_{i_k}^{\mathrm{p}}}{\sum_{k=1}^K s_{i_k}}, \quad \text{where } s_i = \frac{\boldsymbol{e}_i^{\top} \boldsymbol{e}_c}{\|\boldsymbol{e}_i\|\|\boldsymbol{e}_c\|}, \tag{11}$$

where $\boldsymbol{x}_c^{\mathrm{r}}$ is the guidance target for calculating guidance score.

#### 3.4.2 RETRIEVAL GUIDED SCORE ESTIMATION

The objective for retrieval guided sampling is to draw the probability density of predicted forecast distribution $p(\hat{\boldsymbol{x}}^{\mathrm{p}}|\boldsymbol{e})$ closer to the guidance target $\boldsymbol{x}^{\mathrm{r}}$. This can be considered as sampling from a exponential tilted distribution:

$$p_{\theta}(\hat{\boldsymbol{x}}^{\mathrm{p}}|\boldsymbol{e}) = p_{\theta}(\boldsymbol{x}^{\mathrm{p}}|\boldsymbol{e})e^{-\lambda\mathcal{E}(\boldsymbol{x}^{\mathrm{r}}, \boldsymbol{x}^{\mathrm{p}})}, \tag{12}$$

where $\mathcal{E}$ is an energy function. Therefore, the score function for sampling from $p(\hat{\boldsymbol{x}}^{\mathrm{p}}|\boldsymbol{e})$ can be written into:

$$\nabla_{\hat{\boldsymbol{x}}_n^{\mathrm{p}}} \log p_{\theta}(\hat{\boldsymbol{x}}_n^{\mathrm{p}}|\boldsymbol{e}) = \nabla_{\boldsymbol{x}_n^{\mathrm{p}}} \log p_{\theta}(\boldsymbol{x}_n^{\mathrm{p}}|\boldsymbol{e}) - \lambda\nabla_{\boldsymbol{x}_n^{\mathrm{p}}} \mathcal{E}(\boldsymbol{x}^{\mathrm{r}}, \boldsymbol{x}_n^{\mathrm{p}}), \tag{13}$$

where $\lambda$ is a hyperparameter controlling the strength of guidance when modifying the sampling score. The score $\nabla_{\boldsymbol{x}_n^{\mathrm{p}}} \log p_\theta(\boldsymbol{x}_n^{\mathrm{p}}|\boldsymbol{e})$ is approximated by noise prediction $\hat{\boldsymbol{\epsilon}}_n = \boldsymbol{\epsilon}_\theta(\boldsymbol{x}_{n+1}^{\mathrm{p}}, n, \boldsymbol{e})$. The score of energy function is estimated by first approximating a reconstruction $\boldsymbol{x}_n^{\mathrm{p}} = \frac{1}{\sqrt{1-\beta_t}}\left(\boldsymbol{x}_{n+1}^{\mathrm{p}} - \frac{\beta_t}{\sqrt{1-\bar{\alpha}_t}}\hat{\boldsymbol{\epsilon}}\right)$ and then derive the gradient of energy against it. Empirically, we employ L2 distance as the energy function and scale the standard derivation of each energy score estimation to 1. The algorithm pseudo code is presented as Algorithm 1.

---

**Algorithm 1** Sampling with retrieval guidance.

---

**Require:** $\boldsymbol{X}^{\mathrm{o}} \in \mathcal{D}_{\mathrm{test}}$, retrieval database $\mathcal{D}_{\mathrm{retrieval}}$
**Ensure:** Generated time series forecasts $\hat{\boldsymbol{X}}^{\mathrm{p}}$
1: Encode $\boldsymbol{X}^{\mathrm{o}}$ into sampling condition as well as retrieval query $\boldsymbol{e}$
2: Retrieve with Equation (11) to get $\boldsymbol{x}_c^{\mathrm{r}}$.
3: Randomly sample noise $\hat{\boldsymbol{X}}^{\mathrm{p}}{}_N \sim \mathcal{N}(\boldsymbol{0}, \boldsymbol{I})$
4: **for** n from N to 1 **do**
5:     Predict step noise with $\hat{\boldsymbol{\epsilon}}_n = \boldsymbol{\epsilon}_\theta(\boldsymbol{x}_{n+1}^{\mathrm{p}}, n, \boldsymbol{e})$
6:     Tilt the score estimation with Equation (13)
7:     Denoise $\hat{\boldsymbol{X}}^{\mathrm{p}}{}_{n-1} = \frac{\hat{\boldsymbol{X}}^{\mathrm{p}}{}_n - \sqrt{1-\bar{\alpha}_n}\tilde{\boldsymbol{\epsilon}}_n}{\sqrt{\bar{\alpha}_n}}$
8: **end for**
9: $\hat{\boldsymbol{X}}^{\mathrm{p}} = \hat{\boldsymbol{X}}^{\mathrm{p}}{}_0$

---

# 4 EXPERIMENTS

## 4.1 EXPERIMENTAL SETTINGS

### 4.1.1 DATASETS AND BASELINES

We evaluate our approach on four standard time series benchmarks spanning multiple domains: (1) ETT-small (Electricity Transformer Temperature): containing hourly recordings from 2 electricity transformers (**ETTh1**, **ETTh2**) and 15-minute interval data (**ETTm1**, **ETTm2**), each with 7 features; (2) **Electricity**: dataset with hourly consumption of 321 clients; (3) **Traffic**: dataset measuring 15-minute occupancy rates from 862 California highway sensors; (4) **Weather**: dataset with 21 meteorological indicators recorded every 10 minutes throughout 2020.

To demonstrate the effectiveness of our method, we compare with a variety of state-of-the-art time series forecasting methods, on both probabilistic forecasting and point forecasting basis. These include TimeDiff (Shen & Kwok, 2023), TMDM (Li et al., 2024) and NsDiff (Ye et al., 2025) as representative probabilistic forecasting methods, TimesNet (Wu et al., 2023), PatchTST (Nie et al., 2023) , TimeMixer (Wang et al., 2024), TSMixer (Chen et al., 2023), and iTransformer (Liu et al., 2024) represent strong point forecasting methods. For all the baseline methods, we adopt their open-sourced code implementation while keeping the same forecasting setting accross all baselines.

### 4.1.2 EVALUATION METRICS

To access the effectiveness, we assess model performance using four standard metrics: (1) Mean Squared Error (MSE) measuring point prediction accuracy; (2) Mean Absolute Error (MAE) evaluating quantile forecasts with outlier robustness; (3) Continuous Ranked Probability Score (CRPS) quantifying full predictive distribution quality; (4) Quantile Interval Coverage Error (QICE) validating all prediction interval calibration. For deterministic forecasting methods, we report MAE and MSE. For probabilistic methods, we report all metrics statistics collected over 100 generation samples, where MAE and MSE is calculated with the numerical average of all samples. All metrics are averaged across both channels and prediction horizons.

### 4.1.3 IMPLEMENTATION DETAILS

All datasets are split chronologically following common practice, where a 60%:20%:20% split is applied for ETT-small and 70%-10%-20% for all other datasets. We evaluate all methods with the same set of target prediction lengths of {96, 192, 336, 720} for all datasets and baselines. To ensure

fair comparison, we search for the best look back context length among the range of {96, 168, 256, 512} steps. For MiDDiR experiment, the number of retrieved samples $K$ and the retrieval guidance scale $\lambda$ are selected using the combination that produces lowest MAE score in validation set. All experiments are conducted on NVIDIA RTX A6000 GPU with 48GB graphical memory.

## 4.2 EXPERIMENTAL RESULTS

### 4.2.1 PROBABILISTIC FORECASTING

In this experiment, we evaluate the probabilistic forecasting performance of MiDDiR model on the all the datasets, comparing with recent generative forecasting baselines. Table 1 shows the numerical results in CRPS and QICE score. In general, our model achieves the best performance against all baselines, outperforming the second best baselines at a significant margin. On average, our model surpass the second best NsDiff model on CRPS by around 21.9%, and outperform TMDM on QICE by 41.0%. Note that TMDM fails to produce result in traffic dataset due to an OOM error, which represents a high-dimensional setting, implying a high computational resource demand for obtaining satisfying results. In contrast, our MiDDiR model provides the state-of-the-art probabilistic forecasting performance while retaining lower computational requirements.

Table 1: Experiment results on probabilistic forecasting metrics. Best results are highlighted in **bold**. Second best results are underlined. "-" indicates failure in obtaining a result due to OOM error. A lower score indicates better performance.

| Metrics | Model | ETTh1 | ETTh2 | ETTm1 | ETTm2 | Electricity | Traffic | Weather | Average |
|---------|-------|-------|-------|-------|-------|-------------|---------|---------|---------|
| **CRPS** | **TimeDiff** | 0.462 | 0.514 | 0.479 | 0.334 | 0.740 | 0.777 | 0.319 | 0.518 |
| | **TMDM** | 0.459 | 0.358 | 0.373 | 0.320 | 0.310 | - | 0.239 | 0.343 |
| | **NsDiff** | 0.381 | 0.328 | 0.350 | 0.262 | 0.265 | 0.342 | 0.251 | 0.311 |
| | **MiDDiR** | **0.308** | **0.289** | **0.266** | **0.227** | **0.192** | **0.231** | **0.191** | **0.243** |
| **QICE** | **TimeDiff** | 14.817 | 14.877 | 14.829 | 13.374 | 15.493 | 15.426 | 13.726 | 14.649 |
| | **TMDM** | 3.944 | 3.612 | 2.410 | 3.227 | 5.248 | - | 5.158 | 3.933 |
| | **NsDiff** | 1.851 | 2.435 | 3.653 | 3.380 | 6.997 | 8.002 | 4.090 | 4.344 |
| | **MiDDiR** | **1.763** | **2.050** | **1.623** | **2.162** | **1.025** | **4.918** | **2.715** | **2.322** |

### 4.2.2 POINT FORECASTING

In this experiment, we evaluate the point forecasting performance of MiDDiR model, comparing with both generative forecasting baseline and point forecasting baseline. Table 2 shows the evaluation results in MAE and MSE. From Table 2, our model achieves the best average point forecasting performance among the generative forecasting baselines. Although in some of the datasets, there exists a performance gap between probabilistic forecasting models and point forecasting models, our model has achieved the second best place in average among all evaluated baselines, showing its potential in advancing generative time series forecasting model design.

### 4.2.3 EVALUATION ON GIFT-EVAL

We additionally conduct experiments on GIFT-Eval, covering all multivariate tasks with publicly available and reproducible baselines. We report averaged metrics across tasks to reflect robustness over diverse domains, including all multivariate subsets in GIFT-Eval. MiDDiR ranks the 1st on MAPE metric among all non-fundamental deep learning models on medium and long horizon, and ranking the second on short horizon. When compared with all time series models, including all fundamental models, MiDDiR ranked the 3rd on MSE and NMRSE out of all 39 models in comparison. On CRPS metrics, MiDDiR ranked the 10th out of all 39 baselines, which is competitive considering the much lower computational resource consumed for training MiDDiR than those fundamental time series models. These results confirm that MiDDiR's improvements extend beyond the standard datasets.

Table 2: Mean absolute error (MAE) results on point forecasting experiments. Type column indicates the whether the model is a point forecasting method or probabilistic forecasting method in nature. Best results are highlighted in **bold**. Second best results are underlined. A lower score indicates better performance.

| Type | Model | ETTh1 | ETTh2 | ETTm1 | ETTm2 | Electricity | Traffic | Weather | Average |
|------|-------|-------|-------|-------|-------|-------------|---------|---------|---------|
| Point | **PatchTST** | 0.472 | 0.421 | 0.385 | 0.330 | 0.271 | **0.272** | 0.267 | 0.345 |
| | **TSMixer** | 0.585 | 1.032 | 0.452 | 0.812 | 0.306 | 0.377 | 0.279 | 0.549 |
| | **TimeMixer** | **0.436** | 0.399 | 0.385 | 0.318 | **0.268** | 0.284 | 0.265 | 0.336 |
| | **TimesNet** | 0.461 | 0.427 | 0.413 | 0.331 | 0.295 | 0.327 | 0.283 | 0.362 |
| | **iTransformer** | 0.456 | 0.409 | 0.395 | 0.330 | 0.256 | 0.275 | 0.273 | 0.342 |
| Prob. | **TimeDiff** | 0.480 | 0.551 | 0.497 | 0.364 | 0.759 | 0.793 | 0.335 | 0.540 |
| | **NsDiff** | 0.528 | 0.451 | 0.468 | 0.370 | 0.304 | 0.373 | 0.314 | 0.401 |
| | **MiDDiR** | 0.440 | **0.401** | **0.380** | **0.313** | 0.272 | 0.284 | **0.261** | **0.336** |

## 4.3 DISCUSSION ON RETRIEVAL GUIDANCE

In this experiment, we investigate the role of the retrieval guidance technique and investigate its effectiveness, we conduct an ablation study on the guidance strength, by adjusting it in the range of $\lambda = \{0, 0.005, 0.01, 0.02, 0.05\}$, where $\lambda = 0$ represents the case of no guidance. Table 8 shows the effects for adjusting the guidance strength in ETTm1 and Traffic datasets and in the forecasting horizon setting in $96, 192$. From Table 8, we can find that as the guidance scale get larger, the forecasting performance of the model, evaluated by MAE and CRPS, first get better and then increased in ETTm1 dataset. This indicats a potential overfitting problem for high guidance scale that sticking too much to the training dataset where the retrieval database is built upon. However, the phenomenon has not yet emerged for Traffic dataset with the current guidance scale setting. Indicating that retrieval guidance may be more beneficial for the higher dimensional forecasting settings. To

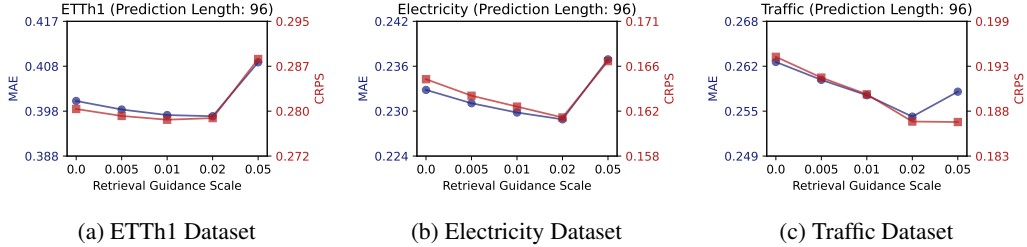

| (a) ETTh1 Dataset | (b) Electricity Dataset | (c) Traffic Dataset |
|---|---|---|

Figure 2: Left axis: MAE. Right axis: CRPS. The x-axis represents different magnitude of retrieval guidance scale.

provide better understanding of the influence of retrieval guidance, we conduct a visualization of retrieval guided samples. From these samples, we can see that the generated time series gradually get closer to the retrieved series and generating a tighter prediction interval, as the guidance scale get larger. This observation confirms the effectiveness of retrieval guidance technique in tilting sampling distribution, validating our design.

Table 3: Ablation study on Retrieval Guidance and Channel Dependent Encoding.

| | ETTh1 | | ETTm1 | | Weather | | Electricity | | Traffic | |
|---|-------|-----|-------|-----|---------|-----|-------------|-----|---------|-----|
| | MAE | CRPS | MAE | CRPS | MAE | CRPS | MAE | CRPS | MAE | CRPS |
| **MiDDiR** | **0.431** | **0.300** | **0.368** | **0.258** | **0.242** | 2.536 | **0.249** | **0.177** | **0.269** | **0.198** |
| **w/o RG** | 0.432 | 0.301 | 0.372 | 0.260 | 0.243 | **2.306** | 0.253 | 0.181 | 0.277 | 0.207 |
| **w/o CD** | 0.434 | 0.300 | 0.379 | 0.266 | 0.242 | 3.767 | 0.254 | 0.179 | 0.298 | 0.231 |
| **w/o CD & RG** | 0.435 | 0.301 | 0.384 | 0.269 | 0.243 | 3.577 | 0.257 | 0.181 | 0.301 | 0.235 |

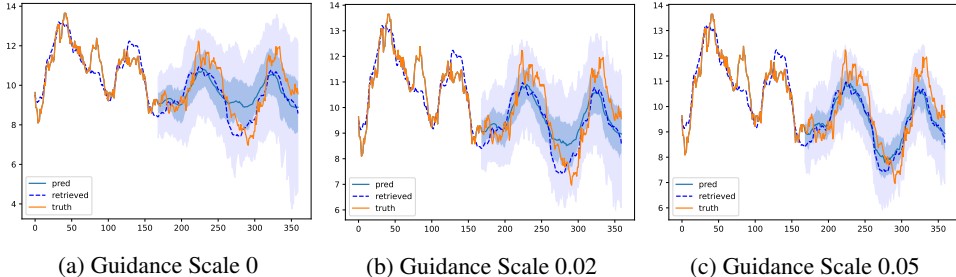

(a) Guidance Scale 0      (b) Guidance Scale 0.02      (c) Guidance Scale 0.05

Figure 3: Retrieval guidance sample with different guidance scale. Orange solid line represents ground truth series. The blue dashed line represents reference series retrieved from training dataset. The blue solid line represents predictions generated by MiDDiR. The shadowed area in blue and purple represents 50% and 90% probabilistic interval, respectively.

### 4.4 INFLUENCE OF MIXED CHANNEL DEPENDENCY

In this experiment, we investigate the effectiveness of mixed channel dependency mechanism in MiDDiR. We replace the channel dependent encoder with a channel independent encoder to construct a channel independent variant of MiDDiR model, notated with "w/o CD". We also examine the result of removing retrieval guidance for this variant, named with "w/o CD & RG". MAE and CRPS results are reported in Table 3, which demonstrate that removing channel dependent encoding produces inferior performance against full MiDDiR model. While retrieval guidance mechanism can lead to enhancement because sequence-level encoding is still informative for retrieval guided forecasting, its benefit is limited compared to the full model with channel dependent encoding, especially on datasets with a large number of channels where an over 50% benefit degradation observed.

Additionally, we look into the attention weight map produced by the attention layer in channel dependent encoder, to discover whether the channels actually rely on each other or not. Figure 4 shows the average attention map of the first layer of a channel dependent encoder for ETTh1 and Weather datasets. Figure 4a demonstrates that a channel dependent pattern is learned by the encoder, while Figure 4b shows a weaker inter-channel correlation that may not produce extra predictive information. This observation corresponds to results in Table 3, indicating that the channel dependent encoder is able to capture a dependency pattern when available, but it retains comparable performance even when only weak channel dependency is learned.

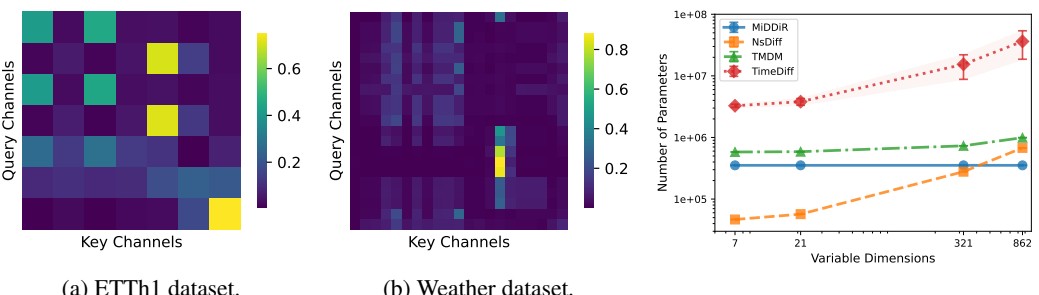

(a) ETTh1 dataset.      (b) Weather dataset.

Figure 4: Attention map by mixed dependency encoder.      Figure 5: Parameter growth.

### 4.5 PARAMETER EFFICIENCY OF CHANNEL INDEPENDENT DECODING

In this experiment, we investigate the parameter efficiency of probabilistic time series forecasting models. We compare the number of trainable parameters among models, in foresting settings that requiring different number of variable dimensions to be modeled. The results are shown in Section 4.4. TimeDiff and TMDM require larger number of parameters than MiDDiR. Although NsDiff model requires fewer parameters when the number of variables or channels is small, it also grows exponentially as the number of variable increases. In contrast, the number of trainable

parameter in MiDDiR is insensitive to the variable channel dimension because all channels are handled independently during probabilistic generation. As a result, MiDDiR presents higher parameter efficiency than baselines with respect to the variable dimension.

## 5 CONCLUSION

In this paper, we propose a mixed channel dependency diffusion model with retrieval guidance (MiD-DiR) to address these challenges. In MiDDiR, we employs a novel mixed channel dependency method on time series diffusion model, encoding historical time series in a channel-dependent manner to obtain informative historical representation while denoising in a channel-independent manner to decrease modeling complexity. During inference, we retrieve similar history occurrence for explicitly tilting the score estimation as retrieval guidance to enhance forecasting quality. Extensive experiments demonstrate the effectiveness of MiDDiR, outperforming baselines in a variety of real-world time series datasets.

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

## LLM USAGE STATEMENT

We utilized large language models (LLMs) to aid in the preparation of this manuscript. Its use was limited to editorial tasks, including proofreading for typographical errors, correcting grammar, and improving the clarity and readability of the text.

## A IMPLEMENTATION DETAILS

### A.1 DENOISING NETWORK ARCHITECTURE AND PARAMETERS

Our channel-independent denoising network is built upon the DiT architecture. Specially for single channel time series, we adopt a patch embedding layer built as 1D-Convolution layer, followed by sequential DiT blocks and a final linear output layer. Each DiT blocks is built with one multi-head self-attention layer and two fully connect layers. Adaptive Layer Normalization is applied to each DiT block and the final layer to integrate historical condition embedding. Specifically, we tune the patch size in {8, 16}, patch embedding dimension to 32, hidden size for MLPs to 64, number of DiT blocks to 4, number of attention heads to 2. Additionally, we apply a parameter dropout rate of 0.1 and denoising condition dropout rate of 0.1 to avoid overfitting. The learning rate is initialized as 1.0e-4 and we adopt a cosine learning rate decay. Training last at most 100 epochs for every datasets, where early stopping is configured with a patience of 10 epochs. The training algorithm is described in Algorithm 2.

## B ADDITIONAL EXPERIMENT RESULTS

We provide the results on MSE metrics, as well as detailed breakdown of experiment results on CRPS, QICE, MAE, MSE metrics.

We also present additional experiment results that corresponds or are complementary to the main results in Section 4.2.

---

**Algorithm 2** Training algorithm

---

**Require:** Time series context $(\boldsymbol{X}^{\mathrm{o}}, \boldsymbol{X}^{\mathrm{p}}) \in \mathcal{D}_{\mathrm{train}}$
**Ensure:** Network parameters $\theta$
1: **repeat**
2:     Sample $(\boldsymbol{X}^{\mathrm{o}}, \boldsymbol{X}^{\mathrm{p}})$ from $\mathcal{D}_{\mathrm{train}}$
3:     Encode $\boldsymbol{X}^{\mathrm{o}}$ with channel dependent encoder with Equations (5) and (6) into channel independent latent
        vector $\boldsymbol{e} = \phi(\boldsymbol{X}^{\mathrm{o}})$
4:     Randomly set $\boldsymbol{e}$ as unconditional identifier $\boldsymbol{e}_u$
5:     Randomly sample time step $n \sim \mathcal{U}(1, N)$
6:     Randomly sample noise $\boldsymbol{\epsilon} \sim \mathcal{N}(\boldsymbol{0}, \boldsymbol{I})$
7:     Corrupt data $\boldsymbol{x}_n = \sqrt{\bar{\alpha}_n}\boldsymbol{x}_0 + \sqrt{1-\bar{\alpha}_n}\boldsymbol{\epsilon}$
8:     Predict step noise with $\tilde{\boldsymbol{\epsilon}} = \boldsymbol{\epsilon}_\theta(\sqrt{\bar{\alpha}_n}\boldsymbol{x}_0^{\mathrm{p}} + \sqrt{1-\bar{\alpha}_n}\boldsymbol{\epsilon}, n, \boldsymbol{e})$
9:     Compute loss with Equation (8) and take gradient step.
10: **until** maximum training step

---

Table 4: Mean squared error (MSE) results on point forecasting experiments. Type column indicates the whether the model is a point forecasting method or probabilistic forecasting method in nature. Best results are highlighted in **bold**. Second best results are underlined. A lower score indicates better performance.

| Type | Model | ETTh1 | ETTh2 | ETTm1 | ETTm2 | Electricity | Traffic | Weather | Average |
|------|-------|-------|-------|-------|-------|-------------|---------|---------|---------|
| | **PatchTST** | 0.484 | 0.396 | **0.354** | 0.271 | **0.172** | **0.433** | 0.229 | 0.334 |
| | **TSMixer** | 0.621 | 1.534 | 0.423 | 1.314 | 0.193 | 0.512 | **0.218** | 0.688 |
| Point | **TimeMixer** | 0.427 | **0.357** | **0.354** | **0.259** | 0.175 | 0.446 | 0.234 | **0.322** |
| | **TimesNet** | 0.455 | 0.392 | 0.393 | 0.281 | 0.194 | 0.614 | 0.248 | 0.368 |
| | **iTransformer** | 0.447 | 0.381 | 0.367 | 0.273 | 0.162 | 0.382 | 0.240 | **0.322** |
| | **TimeDiff** | 0.496 | 0.574 | 0.561 | 0.319 | 0.861 | 1.419 | 0.286 | 0.645 |
| Prob. | **NsDiff** | 0.585 | 0.429 | 0.497 | 0.327 | 0.200 | 0.649 | 0.274 | 0.423 |
| | **MiDDiR** | **0.438** | 0.372 | 0.360 | 0.261 | 0.181 | 0.453 | 0.236 | 0.329 |

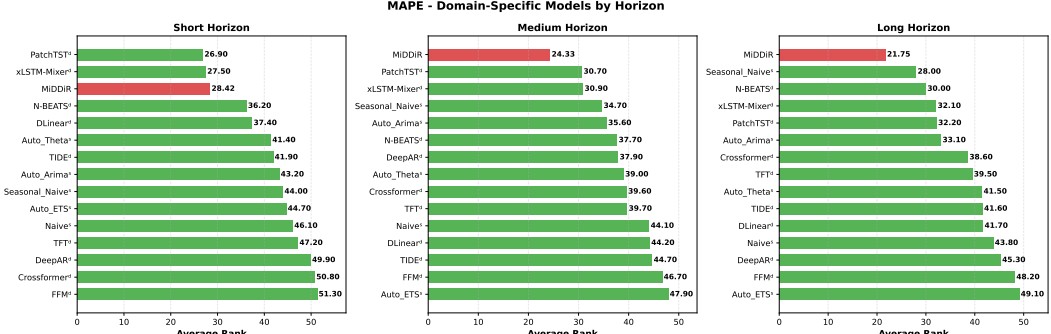

Figure 6: Multi-variate MAPE results on GIFT-Eval, ranked by horizon, for deep learning and statistics models.

## C RESULTS ON GIFT-EVAL

We additionally conduct experiments on GIFT-Eval, covering all multivariate tasks with publicly available and reproducible baselines. We report averaged metrics across tasks to reflect robustness over diverse domains, including all multivariate subsets in GIFT-Eval. As shown in Figure 6, MiDDiR ranks the 1st on MAPE metric among all non-fundamental deep learning models on medium and long horizon, and ranking the second on short horizon. When compared with all time series models, including all fundamental models, MiDDiR ranked the 3rd on MSE and NMRSE out of all 39 models in comparison, as in Figure 7 and Figure 8. As shown in Figure 9, on CRPS metrics, MiDDiR ranked the 10th out of all 39 baselines, which is competitive considering the much lower computational resource consumed for training MiDDiR than those fundamental time series models. These results confirm that MiDDiR's improvements extend beyond the standard datasets.

Table 5: Full experiment results on probabilistic forecasting metrics. Best results are highlighted in **bold**. Second best results are underlined. "-" indicates failure in obtaining a result due to OOM error. A lower score indicates better performance.

| Dataset | $\tau$ | TimeDiff CRPS | TimeDiff QICE | TMDM CRPS | TMDM QICE | NSDiff CRPS | NSDiff QICE | MiDDiR CRPS | MiDDiR QICE |
|---|---|---|---|---|---|---|---|---|---|
| **ETTh1** | 96 | 0.426 | 14.758 | 0.377 | 3.079 | 0.362 | 1.601 | 0.273 | 1.092 |
| | 192 | 0.448 | 14.687 | 0.413 | 3.099 | 0.369 | 1.553 | 0.298 | 1.445 |
| | 336 | 0.472 | 14.869 | 0.519 | 4.568 | 0.379 | 2.068 | 0.315 | 2.371 |
| | 720 | 0.501 | 14.953 | 0.527 | 5.029 | 0.414 | 2.181 | 0.344 | 2.144 |
| | Avg | 0.462 | 14.817 | 0.459 | 3.944 | 0.381 | 1.851 | 0.308 | 1.763 |
| **ETTh2** | 96 | 0.390 | 14.408 | 0.306 | 3.245 | 0.290 | 2.416 | 0.245 | 1.905 |
| | 192 | 0.488 | 14.849 | 0.358 | 3.782 | 0.335 | 2.535 | 0.290 | 1.679 |
| | 336 | 0.423 | 14.851 | 0.367 | 3.547 | 0.336 | 2.173 | 0.314 | 2.860 |
| | 720 | 0.754 | 15.399 | 0.400 | 3.876 | 0.350 | 2.616 | 0.306 | 1.754 |
| | Avg | 0.514 | 14.877 | 0.358 | 3.612 | 0.328 | 2.435 | 0.289 | 2.050 |
| **ETTm1** | 96 | 0.471 | 14.765 | 0.347 | 2.636 | 0.316 | 3.248 | 0.240 | 1.720 |
| | 192 | 0.474 | 14.733 | 0.349 | 1.987 | 0.354 | 4.601 | 0.258 | 1.456 |
| | 336 | 0.479 | 14.883 | 0.376 | 2.359 | 0.349 | 2.654 | 0.270 | 1.316 |
| | 720 | 0.492 | 14.936 | 0.421 | 2.657 | 0.380 | 4.107 | 0.296 | 1.999 |
| | Avg | 0.479 | 14.829 | 0.373 | 2.410 | 0.350 | 3.653 | 0.266 | 1.623 |
| **ETTm2** | 96 | 0.267 | 12.947 | 0.242 | 2.786 | 0.231 | 3.235 | 0.178 | 1.701 |
| | 192 | 0.285 | 13.161 | 0.211 | 1.813 | 0.253 | 3.177 | 0.208 | 1.796 |
| | 336 | 0.347 | 12.923 | 0.307 | 3.152 | 0.234 | 4.001 | 0.237 | 2.214 |
| | 720 | 0.437 | 14.464 | 0.522 | 5.155 | 0.329 | 3.106 | 0.283 | 2.938 |
| | Avg | 0.334 | 13.374 | 0.320 | 3.227 | 0.262 | 3.380 | 0.227 | 2.162 |
| **Electricity** | 96 | 0.734 | 15.451 | 0.330 | 3.810 | 0.269 | 7.563 | 0.162 | 0.784 |
| | 192 | 0.734 | 15.490 | 0.290 | 6.685 | 0.263 | 7.264 | 0.177 | 0.987 |
| | 336 | 0.737 | 15.511 | - | - | 0.259 | 6.660 | 0.187 | 1.371 |
| | 720 | 0.756 | 15.519 | - | - | 0.269 | 6.502 | 0.242 | 0.956 |
| | Avg | 0.740 | 15.493 | 0.310 | 5.248 | 0.265 | 6.997 | 0.192 | 1.025 |
| **Traffic** | 96 | 0.787 | 15.447 | - | - | 0.341 | 8.088 | 0.187 | 1.091 |
| | 192 | 0.778 | 15.427 | - | - | 0.338 | 8.164 | 0.198 | 1.138 |
| | 336 | 0.767 | 15.403 | - | - | 0.341 | 8.066 | 0.203 | 1.441 |
| | 720 | - | - | - | - | 0.347 | 7.689 | 0.337 | 16.000 |
| | Avg | 0.777 | 15.426 | - | - | 0.342 | 8.002 | 0.231 | 4.918 |
| **Weather** | 96 | 0.258 | 13.584 | 0.196 | 4.905 | 0.217 | 4.316 | 0.140 | 2.509 |
| | 192 | 0.303 | 13.594 | 0.240 | 5.850 | 0.234 | 4.141 | 0.173 | 3.186 |
| | 336 | 0.337 | 13.693 | 0.282 | 4.719 | 0.255 | 3.724 | 0.204 | 2.818 |
| | 720 | 0.378 | 14.032 | - | - | 0.299 | 4.177 | 0.247 | 2.347 |
| | Avg | 0.319 | 13.726 | 0.239 | 5.158 | 0.251 | 4.090 | 0.191 | 2.715 |

## D  RETRIEVAL OVERHEAD ANALYSIS

Retrieval is executed only once at the beginning of inference rather than at every diffusion step, making the cost easily amortized. Table 9 summarizes the retrieval overhead across datasets of varying scale. The results show that the average search time per variable is extremely small (0.054–0.176 ms), and even large datasets maintain negligible retrieval latency.

To further quantify runtime impact, we measure the sampling-time overhead of integrating retrieval guidance into diffusion model prediction. As reported in Table 10, a typical diffusion sampling step requires 9–10 ms, while enabling retrieval guidance increases step time by only 0.51%–0.86%, confirming that retrieval introduces only minimal additional cost.

Table 6: Full experiment results on MAE forecasting metrics. Best results are highlighted in **bold**. A lower score indicates better performance.

| Dataset | Length | PatchTST | TSMixer | TimeMixer | TimesNet | iTransformer | TimeDiff | NsDiff | MiDDiR |
|---------|--------|----------|---------|-----------|----------|--------------|----------|--------|--------|
| ETTh1 | 96 | 0.412 | 0.488 | 0.401 | 0.428 | 0.411 | 0.442 | 0.487 | **0.394** |
| | 192 | 0.459 | 0.570 | 0.426 | 0.449 | 0.443 | 0.468 | 0.513 | **0.429** |
| | 336 | 0.459 | 0.613 | **0.442** | 0.477 | 0.457 | 0.491 | 0.530 | 0.458 |
| | 720 | 0.558 | 0.669 | 0.474 | 0.488 | 0.511 | 0.520 | 0.582 | 0.478 |
| | Avg | 0.472 | 0.585 | 0.436 | 0.461 | 0.456 | 0.480 | 0.528 | 0.440 |
| ETTh2 | 96 | 0.362 | 0.833 | **0.346** | 0.377 | 0.353 | 0.406 | 0.420 | 0.337 |
| | 192 | 0.423 | 1.045 | **0.384** | 0.440 | 0.403 | 0.505 | 0.458 | 0.402 |
| | 336 | 0.433 | 1.094 | 0.417 | 0.435 | 0.433 | 0.522 | 0.446 | **0.434** |
| | 720 | 0.465 | 1.157 | 0.448 | 0.456 | 0.446 | 0.770 | 0.479 | 0.429 |
| | Avg | 0.421 | 1.032 | 0.399 | 0.427 | 0.409 | 0.551 | 0.451 | 0.401 |
| ETTm1 | 96 | 0.344 | 0.401 | 0.349 | 0.367 | 0.355 | 0.487 | 0.427 | **0.341** |
| | 192 | 0.374 | 0.425 | 0.370 | 0.400 | 0.383 | 0.495 | 0.458 | **0.368** |
| | 336 | 0.391 | 0.459 | 0.393 | 0.414 | 0.402 | 0.497 | 0.475 | **0.386** |
| | 720 | 0.432 | 0.524 | 0.426 | 0.469 | 0.438 | 0.510 | 0.513 | 0.425 |
| | Avg | 0.385 | 0.452 | 0.385 | 0.413 | 0.395 | 0.497 | 0.468 | 0.380 |
| ETTm2 | 96 | 0.272 | 0.351 | 0.257 | 0.270 | 0.269 | 0.299 | 0.302 | **0.248** |
| | 192 | 0.313 | 0.605 | 0.295 | 0.311 | 0.310 | 0.333 | 0.332 | **0.288** |
| | 336 | 0.339 | 0.765 | 0.333 | 0.342 | 0.345 | 0.370 | 0.411 | **0.326** |
| | 720 | 0.395 | 1.525 | 0.385 | 0.400 | 0.394 | 0.453 | 0.434 | 0.388 |
| | Avg | 0.330 | 0.812 | 0.318 | 0.331 | 0.330 | 0.364 | 0.370 | 0.313 |
| Electricity | 96 | 0.244 | 0.263 | 0.240 | 0.274 | 0.228 | 0.759 | 0.287 | **0.229** |
| | 192 | 0.258 | 0.291 | **0.253** | 0.292 | 0.248 | 0.750 | 0.299 | 0.250 |
| | 336 | 0.276 | 0.325 | 0.273 | 0.299 | 0.264 | 0.754 | 0.309 | **0.263** |
| | 720 | 0.307 | 0.345 | 0.306 | 0.316 | 0.285 | 0.772 | 0.321 | 0.346 |
| | Avg | 0.271 | 0.306 | 0.268 | 0.295 | 0.256 | 0.759 | 0.304 | 0.272 |
| Traffic | 96 | 0.260 | 0.365 | 0.266 | 0.315 | 0.258 | 0.803 | 0.366 | **0.255** |
| | 192 | 0.265 | 0.358 | 0.270 | 0.325 | 0.267 | 0.794 | 0.361 | **0.268** |
| | 336 | 0.271 | 0.382 | 0.290 | 0.330 | **0.274** | 0.783 | 0.374 | 0.276 |
| | 720 | 0.292 | 0.402 | 0.308 | 0.339 | 0.302 | | 0.389 | 0.337 |
| | Avg | 0.272 | 0.377 | 0.284 | 0.327 | 0.275 | 0.793 | 0.373 | 0.284 |
| Weather | 96 | 0.201 | 0.213 | 0.203 | 0.215 | 0.211 | 0.273 | 0.263 | **0.193** |
| | 192 | 0.245 | 0.257 | 0.240 | 0.262 | 0.250 | 0.318 | 0.293 | **0.236** |
| | 336 | 0.287 | 0.296 | 0.280 | 0.303 | 0.288 | 0.353 | 0.324 | 0.278 |
| | 720 | 0.335 | 0.350 | 0.337 | 0.352 | 0.343 | 0.394 | 0.376 | 0.337 |
| | Avg | 0.267 | 0.279 | 0.265 | 0.283 | 0.273 | 0.335 | 0.314 | **0.261** |

## E    RETRIEVAL EFFECTIVENESS AGAINST POTENTIAL DISTRIBUTION SHIFT

We further conduct robustness experiments evaluating the effect of guidance scale $\lambda$ under different density levels, where density is estimated using the average cosine similarity between each test query and its top-10 retrieved items. This similarity/density level also serves as indicator for distribution shift Lower similarity indicates a lower-density region in the data manifold. Using the ETTh1 dataset with a context length of 168 and a prediction length of 192, we divide the test set into three groups: the lowest 25% as low-density, the highest 25% as high-density, and the remaining samples as mid-density.

As shown in Table 11, varying $\lambda$ reveals a consistent performance trend across all density regions, with optimal values around 0.01–0.02. Notably, the improvement from retrieval guidance is most significant in low-density regions, demonstrating both the robustness of the method and its particular strength when forecasting under scarce or weakly correlated historical patterns.

## F    RETRIEVAL ROBUSTNESS OVER VARIOUS SIMILARITY MEASURES

These experiments evaluate both geometric distance metrics and direction-based similarity measures, applied to encoded representations as well as preprocessed raw sequences. All evaluations are performed on the ETTh1 dataset using a context length of 168 and a prediction length of 192, with L2

Table 7: Full experiment results on MSE forecasting metrics. Best results are highlighted in **bold**. A lower score indicates better performance.

| Dataset | Length | PatchTST | TSMixer | TimeMixer | TimesNet | iTransformer | TimeDiff | NsDiff | MiDDiR |
|---|---|---|---|---|---|---|---|---|---|
| ETTh1 | 96 | 0.389 | 0.472 | 0.382 | 0.410 | 0.392 | 0.450 | 0.535 | **0.373** |
| | 192 | 0.446 | 0.599 | 0.411 | 0.449 | 0.430 | 0.498 | 0.560 | **0.432** |
| | 336 | 0.472 | 0.674 | **0.451** | 0.467 | 0.443 | 0.512 | 0.580 | 0.463 |
| | 720 | 0.630 | 0.737 | 0.465 | 0.495 | 0.521 | 0.522 | 0.663 | 0.482 |
| | Avg | 0.484 | 0.621 | 0.427 | 0.455 | 0.447 | 0.496 | 0.585 | 0.438 |
| ETTh2 | 96 | 0.310 | 1.068 | **0.290** | 0.335 | 0.301 | 0.350 | 0.393 | 0.292 |
| | 192 | 0.405 | 1.550 | **0.337** | 0.409 | 0.376 | 0.502 | 0.454 | 0.381 |
| | 336 | 0.414 | 1.680 | 0.384 | 0.393 | 0.421 | 0.513 | 0.399 | **0.419** |
| | 720 | 0.453 | 1.838 | 0.415 | 0.429 | 0.427 | 0.929 | 0.470 | 0.394 |
| | Avg | 0.396 | 1.534 | 0.357 | 0.392 | 0.381 | 0.574 | 0.429 | 0.372 |
| ETTm1 | 96 | 0.292 | 0.344 | 0.292 | 0.318 | 0.306 | 0.547 | 0.417 | **0.298** |
| | 192 | 0.334 | 0.380 | 0.330 | 0.385 | 0.347 | 0.552 | 0.468 | **0.345** |
| | 336 | 0.362 | 0.435 | 0.369 | 0.395 | 0.380 | 0.561 | 0.510 | **0.366** |
| | 720 | 0.428 | 0.531 | 0.425 | 0.475 | 0.435 | 0.585 | 0.593 | 0.432 |
| | Avg | 0.354 | 0.423 | 0.354 | 0.393 | 0.367 | 0.561 | 0.497 | 0.360 |
| ETTm2 | 96 | 0.181 | 0.234 | 0.171 | 0.185 | 0.177 | 0.213 | 0.211 | **0.168** |
| | 192 | 0.233 | 0.576 | 0.231 | 0.243 | 0.241 | 0.276 | 0.261 | **0.223** |
| | 336 | 0.285 | 0.918 | 0.277 | 0.299 | 0.295 | 0.332 | 0.406 | **0.282** |
| | 720 | 0.384 | 3.529 | 0.355 | 0.395 | 0.378 | 0.453 | 0.430 | 0.370 |
| | Avg | 0.271 | 1.314 | 0.259 | 0.281 | 0.273 | 0.319 | 0.327 | 0.261 |
| Electricity | 96 | 0.142 | 0.152 | 0.144 | 0.170 | 0.133 | 0.865 | 0.179 | **0.136** |
| | 192 | 0.157 | 0.176 | **0.163** | 0.190 | 0.153 | 0.839 | 0.196 | 0.159 |
| | 336 | 0.176 | 0.207 | 0.174 | 0.198 | 0.169 | 0.841 | 0.206 | **0.170** |
| | 720 | 0.214 | 0.236 | 0.217 | 0.218 | 0.192 | 0.898 | 0.220 | 0.260 |
| | Avg | 0.172 | 0.193 | 0.175 | 0.194 | 0.162 | 0.861 | 0.200 | 0.181 |
| Traffic | 96 | 0.399 | 0.486 | 0.394 | 0.589 | 0.354 | 1.437 | 0.645 | **0.387** |
| | 192 | 0.415 | 0.485 | 0.446 | 0.604 | 0.370 | 1.413 | 0.629 | **0.416** |
| | 336 | 0.435 | 0.521 | 0.455 | 0.620 | **0.385** | 1.407 | 0.652 | 0.427 |
| | 720 | 0.484 | 0.555 | 0.487 | 0.642 | 0.418 | - | 0.671 | 0.582 |
| | Avg | 0.433 | 0.512 | 0.446 | 0.614 | 0.382 | 1.419 | 0.649 | 0.453 |
| Weather | 96 | 0.150 | 0.146 | 0.157 | 0.158 | 0.162 | 0.202 | 0.208 | **0.153** |
| | 192 | 0.200 | 0.189 | 0.199 | 0.212 | 0.205 | 0.262 | 0.246 | **0.199** |
| | 336 | 0.247 | 0.236 | 0.245 | 0.279 | 0.257 | 0.313 | 0.280 | 0.257 |
| | 720 | 0.317 | 0.302 | 0.336 | 0.342 | 0.337 | 0.365 | 0.362 | 0.336 |
| | Avg | 0.229 | 0.218 | 0.234 | 0.248 | 0.240 | 0.286 | 0.274 | **0.236** |

Table 8: The MAE and CRPS scores on ETTh1, Electricity and Traffic datasets for forecasting horizon in {96,192}, as the guidance scale grows from 0 to 0.05. Lower score indicates better performance. "Retrived Target" represents scores obtain by directing using retrieved historical target as the prediction.

| | ETTm1 | | | | Traffic | | | | Traffic | | | |
|---|---|---|---|---|---|---|---|---|---|---|---|---|
| Horizon | 96 | | 192 | | 96 | | 192 | | 96 | | 192 | |
| Guidance Scale | MAE | CRPS | MAE | CRPS | MAE | CRPS | MAE | CRPS | MAE | CRPS | MAE | CRPS |
| 0 | 0.400 | 0.280 | 0.432 | 0.301 | 0.233 | 0.165 | 0.253 | 0.181 | 0.262 | 0.195 | 0.277 | 0.207 |
| 0.005 | 0.398 | 0.279 | **0.431** | **0.300** | 0.231 | 0.164 | 0.251 | 0.179 | 0.260 | 0.192 | 0.274 | 0.204 |
| 0.01 | 0.397 | **0.278** | 0.431 | 0.300 | 0.230 | 0.163 | 0.250 | 0.178 | 0.258 | 0.190 | 0.272 | 0.202 |
| 0.02 | **0.397** | 0.279 | 0.435 | 0.303 | **0.229** | **0.162** | **0.249** | **0.177** | **0.255** | **0.187** | **0.269** | **0.198** |
| 0.05 | 0.408 | 0.289 | 0.464 | 0.326 | 0.237 | 0.167 | 0.258 | 0.182 | 0.258 | 0.187 | 0.278 | 0.201 |
| Retrieved Target | 0.464 | 0.464 | 0.535 | 0.535 | 0.251 | 0.251 | 0.269 | 0.269 | 0.279 | 0.279 | 0.313 | 0.313 |

chosen as the energy function for retrieval guidance. The quantitative results are presented in Table 12, and the results visualization is in Figure 10.

Empirically, our retrieval method remains robust across all similarity metrics, exhibiting similar trends with respect to the guidance scale $\lambda$. Among all tested measures, cosine similarity in latent space consistently yields the most stable and competitive performance. Distance-based metrics such as L2 and Manhattan perform reasonably well but tend to be slightly more sensitive to noise

Table 9: Retrieval overhead statistics across different datasets. Retrieval is performed only once at the beginning of inference, allowing the cost to be amortized. As shown, retrieval contributes merely 0.054–0.176 ms of time per variable.

| Dataset | ETTh1/ETTh2 | ETTm1/ETTm2 | Weather | Electricity | Traffic |
|---|---|---|---|---|---|
| **# Train Samples** | 8377 | 34297 | 36624 | 18149 | 12017 |
| **# Test Samples** | 2785 | 11425 | 10444 | 5165 | 3413 |
| **# Variables** | 7 | 7 | 21 | 370 | 862 |
| **Retrieval Index Size** | 58464 | 239904 | 768768 | 5824224 | 10344000 |
| **Avg. Search Time / Sample (ms)** | 0.368 | 0.498 | 2.142 | 37.047 | 151.864 |
| **Avg. Search Time / Variable (ms)** | 0.053 | 0.071 | 0.102 | 0.115 | 0.176 |

Table 10: Impact of retrieval guidance on diffusion sampling time.

| Prediction Length | Avg. Step Time (w/o Guidance) | Avg. Step Time (w/ Guidance) | Difference |
|---|---|---|---|
| 96 | 9.618 | 9.667 | 0.51% |
| 192 | 9.804 | 9.859 | 0.57% |
| 336 | 9.893 | 9.979 | 0.86% |

Table 11: Robustness analysis of guidance scale $\lambda$ under different density levels on the ETTh1 dataset (context length = 168, prediction length = 192).

| Density Level / $\lambda$ | 0 | 0.005 | 0.01 | 0.02 | 0.03 | Best Improvement |
|---|---|---|---|---|---|---|
| **Low** | 0.535 | 0.527 | 0.521 | 0.516 | 0.518 | 3.47% |
| **Mid** | 0.405 | 0.402 | 0.400 | 0.401 | 0.409 | 1.36% |
| **High** | 0.415 | 0.411 | 0.408 | 0.408 | 0.414 | 1.72% |

Table 12: Comparison of different similarity measures used for retrieval under guidance scale $\lambda$. The table reports MAE.

| Similarity Measure / $\lambda$ | 0 | 0.005 | 0.01 | 0.02 | 0.03 |
|---|---|---|---|---|---|
| **Cosine** | 0.432 | 0.430 | 0.429 | 0.429 | 0.433 |
| **L2** | 0.432 | 0.430 | 0.429 | 0.431 | 0.436 |
| **Manhattan** | 0.432 | 0.430 | 0.429 | 0.431 | 0.435 |

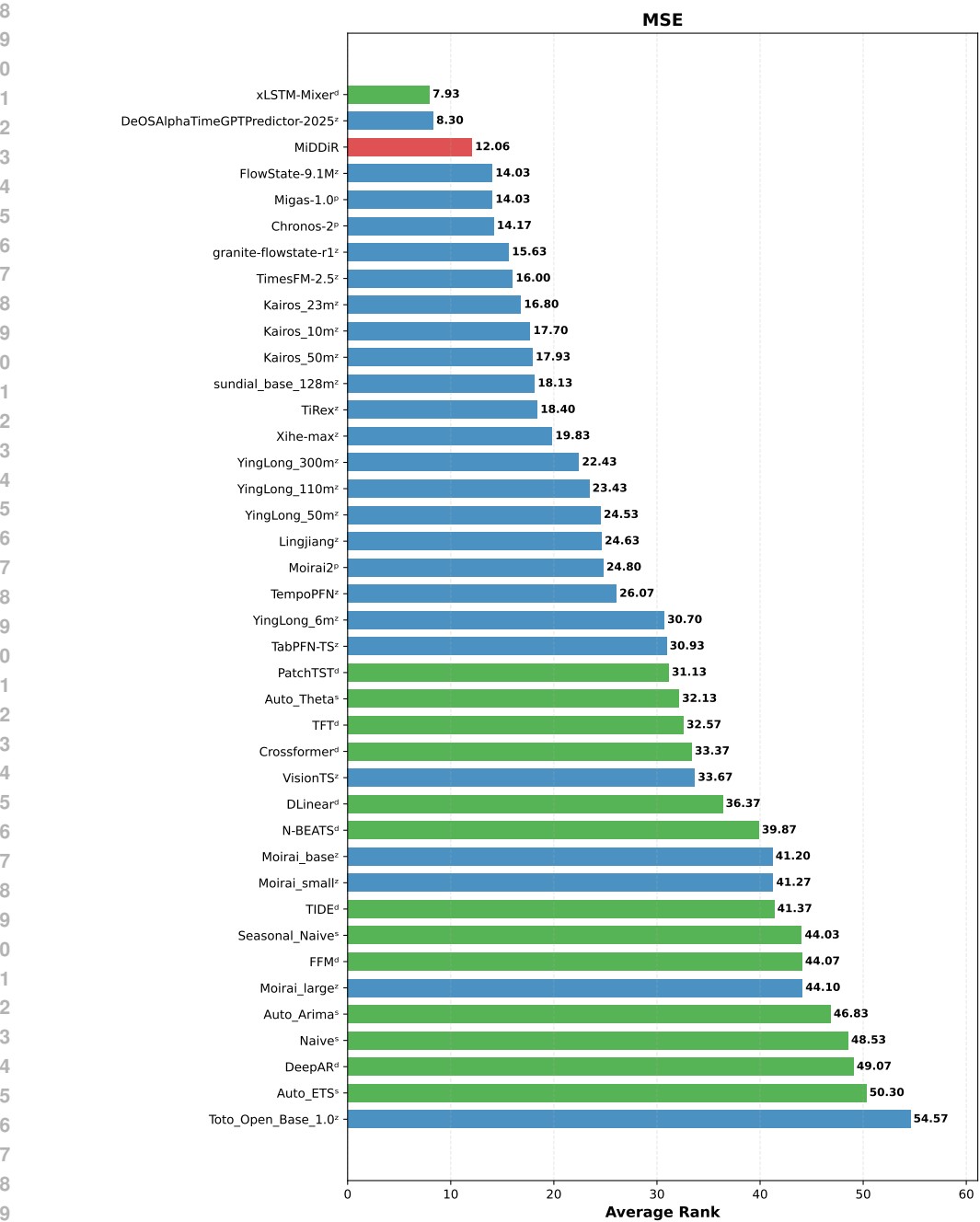

Figure 7: Multi-variate MSE results on GIFT-Eval, ranked for all models.

or local fluctuations in the data distribution. Overall, these findings highlight the flexibility of our retrieval-guided diffusion framework and the particular advantage of cosine similarity when operating in representation space.

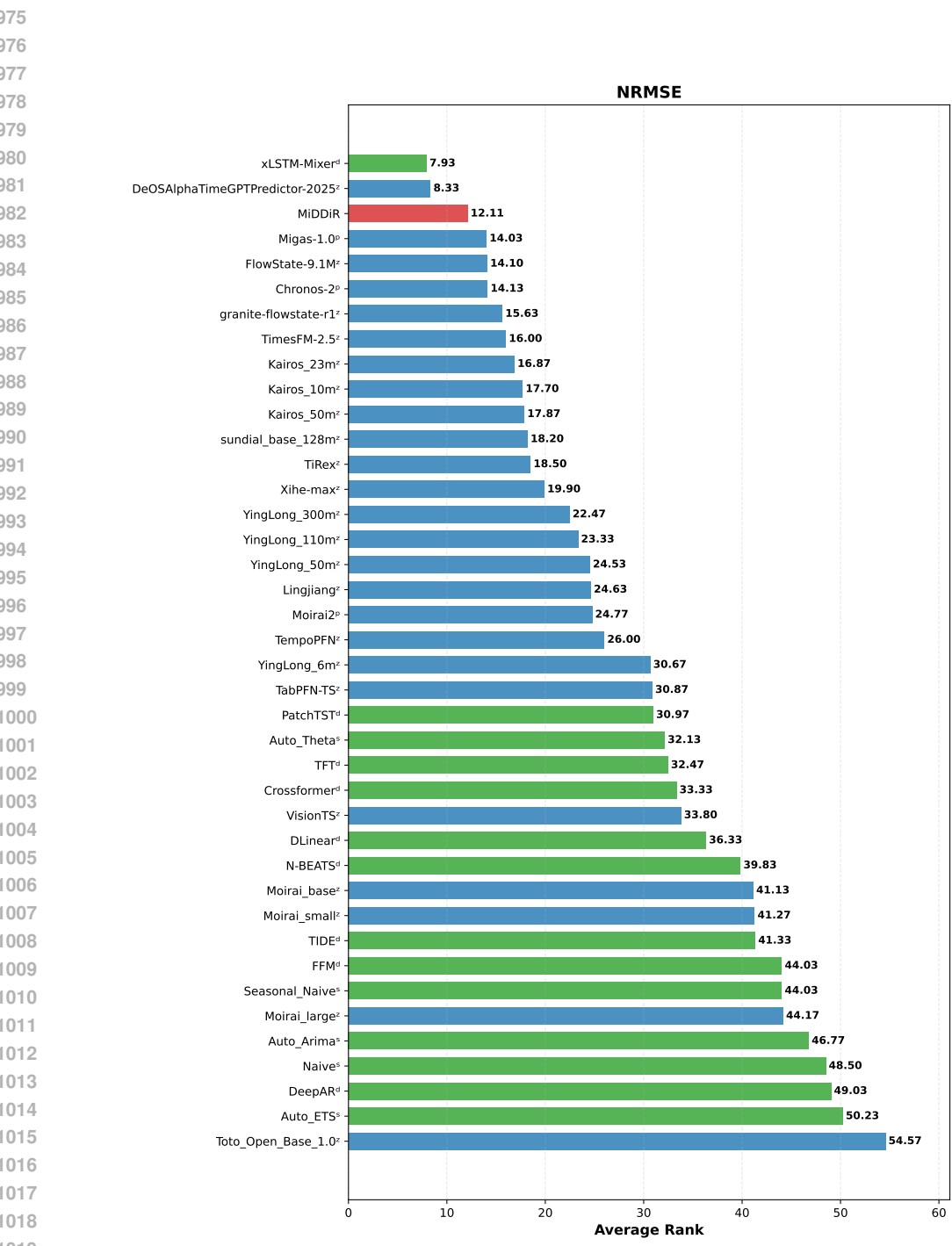

Figure 8: Multi-variate NRMSE results on GIFT-Eval, ranked for all models.

**Quantile_Loss**

| Model | Average Rank |
|---|---|
| DeOSAlphaTimeGPTPredictor-2025[z] | 4.53 |
| Chronos-2[p] | 9.37 |
| TiRex[z] | 11.23 |
| TimesFM-2.5[z] | 11.27 |
| FlowState-9.1M[z] | 12.10 |
| xLSTM-Mixer[d] | 13.77 |
| granite-flowstate-r1[z] | 13.77 |
| Toto_Open_Base_1.0[z] | 14.70 |
| Xihe-max[z] | 15.80 |
| MiDDiR | 19.14 |
| TempoPFN[z] | 20.37 |
| YingLong_300m[z] | 21.47 |
| YingLong_110m[z] | 21.83 |
| Moirai2[p] | 22.27 |
| YingLong_50m[z] | 23.10 |
| Kairos_50m[z] | 23.40 |
| sundial_base_128m[z] | 24.67 |
| TabPFN-TS[z] | 24.70 |
| Kairos_23m[z] | 24.80 |
| Kairos_10m[z] | 25.60 |
| PatchTST[d] | 29.13 |
| Lingjiang[z] | 29.53 |
| YingLong_6m[z] | 30.37 |
| Migas-1.0[p] | 31.67 |
| TFT[d] | 33.70 |
| Moirai_base[z] | 36.07 |
| Moirai_large[z] | 37.50 |
| Moirai_small[z] | 37.63 |
| VisionTS[z] | 41.87 |
| TIDE[d] | 43.27 |
| FFM[d] | 43.73 |
| Auto_Theta[s] | 43.90 |
| Crossformer[d] | 44.47 |
| N-BEATS[d] | 45.17 |
| Seasonal_Naive[s] | 46.33 |
| DLinear[d] | 46.77 |
| DeepAR[d] | 49.07 |
| Auto_Arima[s] | 49.17 |
| Naive[s] | 52.10 |
| Auto_ETS[s] | 52.87 |

Figure 9: Multi-variate CRPS results on GIFT-Eval, ranked for all models.

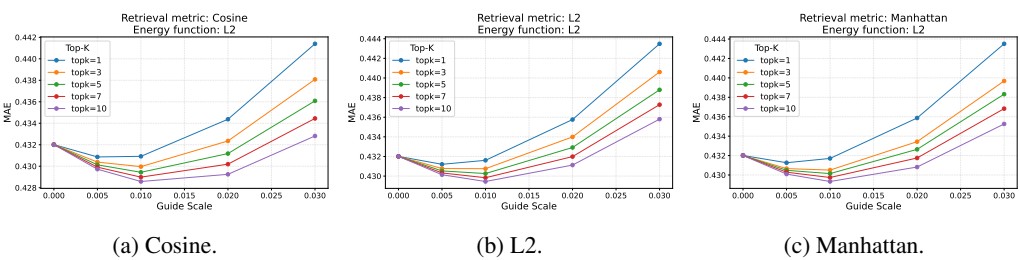

(a) Cosine.    (b) L2.    (c) Manhattan.

Figure 10: Comparison on different retrieval similarity measure.