# OpenReview forum: "Mixed Channel Dependency Diffusion Model with Retrieval Guidance for Time Series Forecasting"
_ICLR.cc/2026/Conference — Submitted to ICLR 2026_

### Official Review · Reviewer_KasR · 2025-10-31

**Soundness:** 2
**Presentation:** 3
**Contribution:** 2
**Rating:** 4
**Confidence:** 3

**Summary:**

This work introduces a promising diffusion-based generative framework that enhances multivariate time series forecasting by addressing challenges of high-dimensional dependency and data sparsity.
The model employs a mixed channel dependency mechanism, where temporal features are encoded with inter-variable awareness while denoising operates independently across channels, balancing expressiveness and scalability.
In addition, a retrieval guidance module retrieves historically similar patterns and integrates them into the diffusion sampling process, effectively improving prediction accuracy in low-density or unseen regions.
Experiments on standard benchmarks (ETT, Electricity, Traffic, and Weather) show that MiDDiR achieves state-of-the-art performance in both deterministic and probabilistic forecasting metrics, outperforming recent diffusion-based models such as NsDiff and TMDM, while offering better parameter efficiency and uncertainty calibration.

**Strengths:**

S1.
MiDDiR introduces a mixed channel dependency diffusion mechanism that combines the strengths of both approaches—encoding inter-channel relationships while performing channel-independent denoising to improve scalability and stability.

S2.
The retrieval-guided diffusion process leverages stored historical patterns to guide sampling in low-density regions of the data manifold.
This idea is novel to the time series domain and represents a creative and transferable paradigm shift, bridging memory-based reasoning with generative diffusion modeling.

S3.
The experimental design is robust: MiDDiR is evaluated on multiple widely used long-horizon benchmarks (ETT, Electricity, Traffic, Weather), using both deterministic metrics (MAE, MSE) and probabilistic scores (CRPS, QICE) to demonstrate consistent improvements.

**Weaknesses:**

W1.
The etrieval guidance might bias generation toward seen patterns rather than truly generalizable dynamics, especially when the data distribution shifts.

W2.
The visualization in Figure 4 suggests varying dependency patterns across datasets, but the analysis stops short of explaining why such differences occur or how they affect model generalization.

W3.
The paper empirically demonstrates the superiority of diffusion-based forecasting but does not explain why the proposed channel-dependent encoding and independent denoising yield better uncertainty calibration or robustness. The theoretical link between these design choices and the underlying data distribution modeling is not clear.

W4.
The experiments rely primarily on standard benchmarks (ETT, Electricity, Traffic, Weather). While these are widely used, they may not sufficiently test the retrieval mechanism’s robustness under different spatiotemporal scales or non-stationarity.
Moreover, the performance on highly dynamic datasets (e.g., finance, healthcare) remains untested.

W5.
The model is evaluated using probabilistic metrics (CRPS, QICE), but there is no analysis of how well the predicted uncertainty intervals reflect true variability.
This omission makes it hard to assess whether MiDDiR provides reliable probabilistic forecasts or merely sharper—but miscalibrated—distributions.

**Questions:**

Q1.
Could you clarify how the mixed channel dependency encoder is trained to balance between inter-channel and intra-channel representations?
For instance, is there an explicit regularization term or architectural design that controls this mixture, or is it purely learned through backpropagation?

Q2.
The paper motivates channel-independent denoising as a way to improve efficiency and reduce parameter coupling, but could the authors provide a theoretical or empirical justification for why independence at the denoising stage does not degrade the learned dependency structure?

Q3.
How does MiDDiR handle distribution shift or noisy retrievals (e.g., when retrieved samples differ significantly from the target input)? Is there a threshold or adaptive weighting mechanism to prevent harmful guidance?

---

> ### Author Response · Authors · 2025-12-03
>
> We sincerely thank the reviewer for the thoughtful and encouraging comprehensive analysis of our work. Below, we provide detailed responses to all questions and concerns.
>
> **W1&Q3. Robustness against distribution shift for retrieval guidance**
>
> We use density of test set samples in the training set density as proxy for investigating distribution shift. We conducted additional robustness experiments of guidance scale $\lambda$ against different density level, where density is approximated by the average cosine similarity between test query and the top 10 matched indexes. A lower cosine similarity between query and best matched dataset sample indicates a lower density and stronger distribution shift. The experiment is conducted on ETTh1 dataset with context length 168 and prediction length 192. We classify the lowest 25% from the test dataset as low-density, the highest 25% as high-density, and all the remaining as mid-density.
>
> The MAE results are shown in the table below. Varying $\lambda$ shows clear and same performance pattern across different density level, with optimal values around 0.01-0.02. The performance improvement brought by retrieval guidance on low-density region is the highest, showing both the robustness and advantage of applying retrieval guidance in low-density patterns.
>
> | **Density level / Guidance scale** | **0** | **0.005** | **0.01** | **0.02** | **0.03** | **Best Improvement** |
> | --- | --- | --- | --- | --- | --- | --- |
> | **Low** | 0.535  | 0.527  | 0.521  | 0.516  | 0.518  | 3.47% |
> | **Mid** | 0.405  | 0.402  | 0.400  | 0.401  | 0.409  | 1.36% |
> | **High** | 0.415  | 0.411  | 0.408  | 0.408  | 0.414  | 1.72% |
>
> **W2. Further explanation for the visualization in Figure 4**
>
> We appreciate the reviewer’s feedback. Fig. 4 is intended to qualitatively illustrate how channel dependencies are captured by our encoder. Rather than showing a uniform diagonal-dominant structure, the attention maps reveal heterogeneous dependency patterns across variables.
>
> Specifically, some variables exhibit near-independence from the others—for example, the last channel in ETTh1 and the 15th channel in Weather, both of which show attention concentrated almost entirely on themselves. In contrast, other variables demonstrate strong cross-channel dependencies: in ETTh1, the 1st and 3rd channels attend strongly to each other, while the 2nd and 4th channels both rely heavily on information from the 5th channel. These non-trivial patterns reflect meaningful inter-variable relationships and match the intuition that certain variables act as key drivers in multivariate dynamics.
>
> **W3. Explanation on why the proposed channel-dependent encoding and independent denoising yield better uncertainty calibration or robustness**
>
> Building on the analysis in our response to W2, the encoder learns cross-channel interactions when they truly exist, while the channel-independent denoising module prevents unnecessary parameter entanglement and thus improves robustness. This division allows the model to capture structured dependencies while avoiding overfitting to spurious ones, which empirically leads to better calibration. The combination provides a more stable decomposition of dependency learning, contributing to improved uncertainty estimation.
>
> **W4. Requires extra benchmark, baselines and highly dynamic datasets**
>
> We appreciate the suggestion regarding dataset diversity. In addition to the standard benchmarks, we now include results on GIFT-eval, which involves more complex and dynamic temporal patterns. To be more specific, MiDDiR ranks the 1st on MAPE metric among all non-fundamental deep learning models on medium and long horizon, and ranking the second on short horizon, confirming that the method generalizes well beyond the previous datasets. When compared with all time series models, including all fundamental models, MiDDiR ranked the 3rd on MSE and NMRSE out of all 39 models in comparison. On CRPS metrics, MiDDiR ranked the 10th out of all 39 baselines, which is competitive considering the much lower computational resource consumed for training MiDDiR than those fundamental time series models. These results confirm that MiDDiR’s improvements extend beyond the previous datasets.
>
> Regarding highly dynamic dataset, we would like to note that the Weather dataset can represent a highly dynamic and non-stationary dataset, and MiDDiR performs strongly in this dataset. The extension into finance/healthcare dataset can be considered as potential future work.

---

> ### Author Response · Authors · 2025-12-03
>
> **W5. Requires analysis of how well the predicted uncertainty intervals reflect true variability**
>
> While we report probabilistic metrics such as CRPS and QICE, we agree that inspecting uncertainty quality provides additional clarity. Section 4.3 and Figure 3 can serve as this required analysis, where we have shown that with a larger guidance scale, the forecasting uncertainty shrinks towards the retrieved sample and thereby result in a distribution that has a higher probability density to be sampled from retrieved distribution.
>
> **Q1. Clarify how the mixed channel dependency encoder is trained to balance between inter-channel and intra-channel representations**
>
> The attention-based channel mixing mechanism in the encoder architecture provide flexibility for the inter-channel and intra-channel balance, where the attention module is learned through back-propagation.
>
> **Q2. Theoretical or empirical justification for why independence at the denoising stage does not degrade the learned dependency structure**
>
> Although denoising is parameterized per channel, it always conditions on the latent representation produced by the shared encoder. The decoder acts as a powerful conditional output head that reconstructs each channel based on this latent vector, which already embeds dependency information. Thus, independence in parameterization does not imply independence in modeling and does not degrade cross-channel structure. Empirical indications can be found in Section 4.4, Table 3, where the channel-dependent variant of our model indeed fails to capture the ideal cross-channel distribution, resulting in inferior forecasting performance.
>
> ---
>
> We sincerely thank the reviewer again for the detailed feedback and encouraging evaluation. Your comments have helped us substantially improve the clarity, completeness, and positioning of our work. We hope our responses adequately address all concerns.

---

### Official Review · Reviewer_RKCx · 2025-10-31

**Soundness:** 2
**Presentation:** 3
**Contribution:** 2
**Rating:** 4
**Confidence:** 4

**Summary:**

This paper introduces MiDDiR, a diffusion model for multivariate time series forecasting. Its core ideas are a "mixed channel dependency" strategy that uses a channel-dependent encoder and a channel-independent denoiser, and a "retrieval guidance" mechanism that leverages similar historical patterns during inference to tilt the sampling process. The authors demonstrate state-of-the-art performance on several benchmarks compared to existing probabilistic and deterministic models.

**Strengths:**

The proposed "mixed channel dependency" is a thoughtful and well-motivated design. It cleverly balances the need to capture informative inter-channel dependencies in the historical context (via the encoder) with the computational benefits of channel-independent generation (via the denoiser). This approach directly tackles the known issue of maximum likelihood training underestimating low-density regions.

**Weaknesses:**

1. Insufficient theoretical explanation and details for the mixed channel dependency mechanism. Although the authors propose the "channel-dependent encoding + channel-independent denoising" architecture, they do not deeply explain the rationality of the connection between the two phases. For example, how is the cross-channel correlation information generated by channel-dependent encoding effectively utilized in channel-independent denoising? If the denoising phase processes each channel completely independently, is there information loss in the cross-channel information captured during the encoding phase? Additionally, details of the channel-dependent encoder design (e.g., basis for selecting the number of attention heads and layers) are not clearly stated.

2. Lack of justification for key design choices in retrieval guidance. The retrieval phase adopts a strategy of "cosine similarity + Top-K weighted average," but the logic for selecting the K value is not explained (only mentioning "selected via the validation set" without providing sensitivity analysis for different K values); meanwhile, the reason for choosing L2 distance as the energy function is not justified, nor is it compared with other distance metrics (e.g., Manhattan distance, Dynamic Time Warping (DTW)—a commonly used metric for time series data), making it impossible to verify the optimality of the current choice.

3. Missing comparison with non-diffusion probabilistic models.

4. Unverified the Performance in longer forecasting scenarios (e.g., 720 steps).

**Questions:**

1. Through what specific means is the cross-channel correlation information captured by the channel-dependent encoder injected into the channel-independent denoising phase?

2. The retrieval database is built based on the training set, and the impact of training set distribution shifts on retrieval effectiveness (e.g., whether retrieval guidance fails when the test set contains new patterns not present in the training set) is not discussed.

---

> ### Author Response · Authors · 2025-12-03
>
> We sincerely thank the reviewer for the thoughtful and encouraging comprehensive analysis of our work. Below, we provide detailed responses to all questions and concerns.
>
> **W1-1&Q1. Rationality of the connection between the channel-dependent encoding and channel-independent denoising**
>
> Our denoising is “channel-independent” only in parameterization, not in conditioning.
> The encoder produces per-channel latent vectors that already embed cross-channel correlations through attention mixing. These vectors are injected into every DiT block via adaptive LayerNorm, so denoising for each channel is always conditioned on multi-channel information. Ablations (“w/o CD”) in Table 3 show clear degradation, confirming that the encoded cross-channel information is effectively used.
>
> In channel-dependent encoder, the output latent representation of each channel comes from the weighted combination of intermediate channel representations, thereby carrying multi-channel information needed as learned by gradient descent. Directly using multiple channels where these channels are not really correlated might inject noise into predictions. Therefore, the encoding phase introduces information selection instead of information loss.
>
> **W1-2. Details of the channel-dependent encoder design**
>
> We have depicted the design in Section 3.2.1. The encoder uses a lightweight, fixed architecture: 2 layers and a single attention head. To keep the mixed-dependency mechanism simple yet effective, we fixed the design and didn’t conduct any architecture search.
>
> **W2-1. Logic for selecting the K value and sensitivity analysis**
>
> We select K by evaluating the validation-set MAE across different K values. We report the consistency between validation and test performance, as well as the sensitivity to different choices of K. These results show that the method perform consistently with varying choice of K, both of which obtain the best result on a K of 10 and guidance scale of 0.01.
>
> Validation MAE across different K and guidance scales:
>
> | **Top-K / Guidance scale** | **0** | **0.005** | **0.01** | **0.02** | **0.03** |
> | --- | --- | --- | --- | --- | --- |
> | **1** | 0.672  | 0.671  | 0.673  | 0.681  | 0.694  |
> | **3** | 0.672  | 0.670  | 0.671  | 0.677  | 0.688  |
> | **5** | 0.672  | 0.669  | 0.669  | 0.675  | 0.684  |
> | **7** | 0.672  | 0.669  | 0.669  | 0.674  | 0.682  |
> | **10** | 0.672  | 0.669  | 0.668  | 0.672  | 0.678  |
>
> Test MAE across different K and guidance scales:
>
> | **Top-K / Guidance scale** | **0** | **0.005** | **0.01** | **0.02** | **0.03** |
> | --- | --- | --- | --- | --- | --- |
> | **1** | 0.432 | 0.431 | 0.431 | 0.434 | 0.441 |
> | **3** | 0.432 | 0.430 | 0.430 | 0.432 | 0.438 |
> | **5** | 0.432 | 0.430 | 0.429 | 0.431 | 0.436 |
> | **7** | 0.432 | 0.430 | 0.429 | 0.430 | 0.434 |
> | **10** | 0.432 | 0.430 | 0.429 | 0.429 | 0.433 |
>
> **W2-2. Selection of energy function**
>
> We additionally experimented with multiple alternative energy functions. The experiments are conducted on ETTh1 dataset with context length 168 and prediction length 192, using cosine similarity as the measure for retrieval. We report the corresponding MAE results. The effectiveness pattern remains stable and effective across all energy functions.
>
> | **Energy function / Guidance Scale** | **0** | **0.005** |  **0.01** |  **0.02** |  **0.03** |
> | --- | --- | --- | --- | --- | --- |
> | **L2** | 0.432 | 0.430 | 0.429 | 0.429 | 0.433 |
> | **MAE** | 0.432 | 0.430 | 0.429 | 0.429 | 0.430 |
> | **Pearson** | 0.432 | 0.431 | 0.430 | 0.430 | 0.433 |
> | **Quantile** | 0.432  | 0.430  | 0.429  | 0.428  | 0.430  |
>
> **W3. Extended experiments comparing non-diffusion probabilistic models**
>
> We have now included a comparisons with non-diffusion probabilistic models on GIFT-Eval. MiDDiR achieves competitive or superior probabilistic performance in these comparisons. To be more specific, MiDDiR ranks the 1st on MAPE metric among all non-fundamental deep learning models on medium and long horizon, and ranking the second on short horizon, confirming that the method generalizes well beyond the previous datasets. When compared with all time series models, including all fundamental models, MiDDiR ranked the 3rd on MSE and NMRSE out of all 39 models in comparison. On CRPS metrics, MiDDiR ranked the 10th out of all 39 baselines, which is competitive considering the much lower computational resource consumed for training MiDDiR than those fundamental time series models. These results confirm that MiDDiR’s improvements extend beyond the previous datasets.
>
> **W4. Extended experiments on longer forecasting setting**
>
> We have conducted additional experiments on longer forecasting horizons and now results have included 720-step prediction. The detailed comparison of different forecast length is depicted in the appendix. The model maintains strong performance even under this extended setting.

---

> ### Author Response · Authors · 2025-12-03
>
> **Q2. Impact of training set distribution shifts on retrieval effectiveness**
>
> We conducted additional robustness experiments of retrieval guidance against different pattern similarity level between test samples and the training set, where similarity level serves as indicator for distribution shift and is measured by the average cosine similarity between test query and the top 10 matched indexes. The experiment is conducted on ETTh1 dataset with context length 168 and prediction length 192. We classify the lowest 25% similarity from the test dataset as “low”, the highest 25% as “high”, and all the remaining as “mid”.
>
> The MAE results are shown in the table below. The performance improvement brought by retrieval guidance on low-similarity region is not degraded, showing both the robustness and advantage of applying retrieval guidance technique.
>
> | **Similarity level / Guidance scale** | **0** | **0.005** | **0.01** | **0.02** | **0.03** | **Best Improvement** |
> | --- | --- | --- | --- | --- | --- | --- |
> | **Low** | 0.535  | 0.527  | 0.521  | 0.516  | 0.518  | 3.47% |
> | **Mid** | 0.405  | 0.402  | 0.400  | 0.401  | 0.409  | 1.36% |
> | **High** | 0.415  | 0.411  | 0.408  | 0.408  | 0.414  | 1.72% |
>
> ---
>
> We sincerely thank the reviewer again for the detailed feedback and encouraging evaluation. Your comments have helped us substantially improve the clarity, completeness, and positioning of our work. We hope our responses adequately address all concerns.

---

### Official Review · Reviewer_HHK8 · 2025-11-08

**Soundness:** 2
**Presentation:** 2
**Contribution:** 1
**Rating:** 4
**Confidence:** 4

**Summary:**

The paper proposes MiDDiR, short for mixed channel dependency diffusion model, for time series forecasting. The paper is motivated by the idea of improving long-horizon forecasts for diffusion-style generative models. MiDDiR has a channel-dependent encoding MLP block which mixes information across different variates in a multivariate time series. This encoding is passed on to a Diffusion Transformer (DiT)-based architecture for denoising and sampling. MiDDiR also allows a mechanism to potentially improve the forecast samples by using the training corpus for additional guidance. This is achieved by selecting related time series in the embedding space and modifying the sampling score function with a weighted guidance term. Experiments on 7 benchmark datasets show that MiDDiR improves on probabilistic forecasting compared to other baselines.

**Strengths:**

- The most interesting aspect about this paper is the idea of retrieval guidance. It could potentially be helpful in specific forecasting scenarios such as cold-start forecasting. However, this specific dimension has not been explored.
- The paper is generally easy to follow.

**Weaknesses:**

- Upon reading the introduction, it is not immediately obvious what the key motivating factor of this work is. As per my understanding, the main motivation is to improve the performance of diffusion-based generative models on long-horizon forecasting. Although there have been several works on the problem of long-horizon forecasting and the long-term forecasting benchmark, it holds limited value in practice.
- "To ensure fair comparison, we 218 evaluate with a look back window of 168 steps and target window in 96, 192, 336 for all datasets." Restricting context length is not ensuring "fair comparison". Some models work better with longer context lengths and practically context length is not restricted and longer contexts are used whenever possible. A fair comparison would be to either provide longer contexts for model that work well with them or experiment with different context lengths and report the best results for each model.
- The evaluation benchmark used in this paper is often criticized for its limitations. 4 of the 7 datasets are essentially the same dataset (ETTh1, ETTh2, ETTm1, ETTm2). Please refer to the talk (and paper) from C. Bergmeir [1, 2] where he discusses the limitation of this benchmark and current evaluation practices. A recent position paper [3] also conducted a comprehensive evaluation of models on this benchmark showing that there's no obvious winner. Authors should consider using better benchmarks to demonstrate the effectiveness of their method. See, for example,
    - Chronos Benchmark II: This benchmark includes 27 datasets (42, if you include Benchmark I) providing a comprehensive coverage over domains, frequencies and other properties [4].
    - GIFT-Eval: This benchmark includes 90+ tasks across multiple datasets and domains. Please refer to https://github.com/SalesforceAIResearch/gift-eval.
    - The Monash Benchmark: https://forecastingdata.org/

[1] https://neurips.cc/virtual/2024/workshop/84712#collapse108471
[2] Hewamalage, Hansika, Klaus Ackermann, and Christoph Bergmeir. "Forecast evaluation for data scientists: common pitfalls and best practices." Data Mining and Knowledge Discovery 37.2 (2023): 788-832.
[3] Brigato, Lorenzo, et al. "Position: There are no Champions in Long-Term Time Series Forecasting." arXiv preprint arXiv:2502.14045 (2025).
[4] Ansari, Abdul Fatir, et al. "Chronos: Learning the language of time series." arXiv preprint arXiv:2403.07815 (2024).

**Questions:**

I previously reviewed this paper for NeurIPS. The results reported for MiDDiR have improved considerably compared to the NeurIPS submission. Can the authors clarify what lead to this improvement?

---

> ### Author Response · Authors · 2025-12-03
>
> We sincerely thank the reviewer for the thoughtful and constructive feedback. We appreciate the helpful suggestions for additional analyses and comparisons. Below we address the reviewer’s concerns in detail.
>
> **W1. Motivation factor and the value of long-horizon time series forecasting**
> We appreciate the reviewer’s concern regarding the motivation. Long-horizon forecasting remains important in many real-world scenarios—such as energy planning, supply-chain management, and environmental monitoring—where decisions must be made well ahead of time and short-term forecasts offer limited actionable insight. From a modeling perspective, long-horizon *multivariate* forecasting requires estimating a high-dimensional conditional distribution, which is substantially more challenging than short-term regression.
>
> Our motivation is therefore twofold:
>
> (i) leverage diffusion models as strong probabilistic estimators for structured long-range generation, and
>
> (ii) address their difficulty in modeling high-dimensional joint dynamics through mixed channel dependency and retrieval-guided alignment.
>
> The proposed channel-dependent encoder extracts cross-variable relationships when they are informative, while the channel-independent denoiser reduces modeling complexity. Retrieval guidance further injects statistically similar historical patterns to stabilize long-range predictions. We will clarify this motivation more explicitly in the revised introduction.
>
> **W2. Extended experiments with different context length**
>
> We agree that restricting context length does not constitute a fair comparison. In the revised experiments, we evaluate all baselines and MiDDiR under multiple context lengths {96,168,256,512} for each forecast horizon {96,192,336,720} and we now report the best performance achieved by each method for each forecast horizon in our main experiment (Section 4.2, Table 1, Table 2). The performance breakdown by forecast horizon in the appendix is also updated. This ensures that models that benefit from longer contexts are evaluated under their preferred settings.
>
> **W3. Extended experiments on extensive benchmarks**
>
> We share the reviewer's concern about the limitations of the long-term forecasting benchmark. To address this, we have conducted additional experiments on GIFT-Eval, covering all multivariate tasks with publicly available and reproducible baselines. We report averaged metrics across tasks to reflect robustness over diverse domains, including all multivariate subsets in GIFT-Eval. MiDDiR ranks the 1st on MAPE metric among all non-fundamental deep learning models on medium and long horizon, and ranking the second on short horizon, confirming that the method generalizes well beyond the previous datasets. When compared with all time series models, including all fundamental models, MiDDiR ranked the 3rd on MSE and NMRSE out of all 39 models in comparison. On CRPS metrics, MiDDiR ranked the 10th out of all 39 baselines, which is competitive considering the much lower computational resource consumed for training MiDDiR than those fundamental time series models. These results confirm that MiDDiR’s improvements extend beyond the previous datasets.
>
> We have incorporated these extended benchmarks in the revised manuscript.
>
> **Q1. Improvements compared to the NeurIPS submission version**
> There are two major factors contributed to the improved performance:
>
> **1. Model revision.**
>
> The channel-dependent encoder has been redesigned to include an attention-based channel-mixing block, yielding more expressive latent representations and more reliable channel-wise retrieval.
>
> **2. Improved experimental protocol.**
>
> We now perform a search over multiple context lengths for all baselines, using a unified evaluation pipeline, and include comprehensive GIFT-eval benchmarks. This leads to more stable and competitive results across datasets. Additionally, we strengthened experimental analysis with ablation studies and sensitivity analysis.
>
> These updates account for the differences relative to the NeurIPS version.
>
> ---
>
> We sincerely thank the reviewer again for the thoughtful comments and constructive suggestions. Hope these clarifications have resolve all concerns.

---

### Official Review · Reviewer_3zLo · 2025-11-10

**Soundness:** 3
**Presentation:** 3
**Contribution:** 3
**Rating:** 6
**Confidence:** 4

**Summary:**

This paper proposes MiDDiR for multivariate time series forecasting. The key innovation is a "mixed channel dependency" strategy that encodes historical data in a channel-dependent manner while performing channel-independent denoising to balance modeling expressiveness and computational efficiency. Additionally, the paper introduces retrieval guidance that tilts the diffusion sampling process using similar historical patterns from the training set. Extensive experiments on multiple real-world datasets demonstrate that MiDDiR outperforms existing baselines in both probabilistic and point forecasting tasks.

**Strengths:**

1.The proposed "mixed channel dependency" strategy leverages channel-dependent encoding to capture inter-channel correlations while maintaining a channel-independent generation approach to reduce computational complexity, representing a well-balanced trade-off for high-dimensional multivariate forecasting.

2.The proposed retrieval-guided sampling method enhances generation quality by leveraging historical patterns, specifically addressing the low-density region problem.

3.The method shows consistent improvements across multiple datasets, achieving substantial gains over baselines

**Weaknesses:**

1.The Retrieval Guidance mechanism drastically increases computational overhead and latency by requiring an expensive Top-K search and calculation at every diffusion step for every channel. The paper fails to provide any quantitative analysis of this critical increase in inference time, making the model's practical utility for real-time long-term forecasting questionable.

2.The retrieval system risks overfitting to the training data and lacks robustness. Over-reliance on the database can be induced by a high guidance strength (λ), while for novel or low-density data patterns, retrieved historical samples may be irrelevant, noisy, or actively misguide the diffusion process, thus degrading prediction quality.

3.The model relies on simple cosine similarity to measure feature similarity during retrieval, a geometric matching approach that may not fully capture complex temporal characteristics such as dynamic patterns, seasonality, or phase alignment in time series data, thereby raising concerns about the robustness of the retrieval process.

4.The paper does not discuss how to efficiently and dynamically update or manage this large-scale retrieval database in a practical online forecasting setting. The lack of consideration for dynamic data environments limits its feasibility for long-term deployment in the real world.

**Questions:**

1.Please quantify the inference time overhead introduced by the retrieval guidance mechanism and discuss potential strategies to reduce this latency.

2.Please include a robustness analysis under different guidance strengths (λ), especially for novel or low-density patterns where retrieval may be unreliable.

3.Consider evaluating more temporally-aware similarity metrics (e.g., DTW or phase-aware measures) to better capture complex time-series dynamics.

4.Please propose a feasible strategy for dynamically updating the retrieval database in evolving data environments to support long-term deployment.

---

> ### Author Response · Authors · 2025-12-03
>
> We sincerely thank the reviewer for the thoughtful and constructive feedback. We appreciate the helpful suggestions for additional analyses and comparisons. Below we address the reviewer’s concerns in detail.
>
> **W1&Q1. Inference time overhead analysis and potential strategies to reduce latency**
>
> We thank the reviewer for the suggestion. We would like to clarify that retrieval is executed only once at the beginning of inference, not at every diffusion step, so the cost can be amortized. We have measured the actual overhead of retrieval on different dataset size, and the results are attached below. As shown in the Table R1, retrieval accounts for 0.054-0.176 ms of total inference time. We have also measured the actual time overhead of diffusion model sampling on various prediction lengths on ETTh1, as shown in Table R2, where a sampling step typically costs 9-10 ms, and enabling retrieval guidance only accounts for a modest 0.51%-0.86% overall increase in sampling step time.
>
> Table R1: Retrieval time overhead analysis
>
> | **Dataset** | **ETTh1/ETTh2** | **ETTm1/ETTm2** | **Weather** | **Electricity** | **Traffic** |
> | --- | --- | --- | --- | --- | --- |
> | **# Train Samples** | 8377 | 34297 | 36624 | 18149 | 12017 |
> | **# Test Samples** | 2785 | 11425 | 10444 | 5165 | 3413 |
> | **# Variables** | 7 | 7 | 21 | 370 | 862 |
> | **Retrieval Index Size** | 58464 | 239904 | 768768 | 5824224 | 10344000 |
> | **Average Search Time Per Sample (ms)** | 0.368 | 0.498 | 2.142 | 37.047 | 151.864 |
> | **Average Search Time Per Variable (ms)** | 0.053 | 0.071 | 0.102 | 0.115 | 0.176 |
>
> Table R2: Guided sampling time overhead analysis
>
> | **Prediction Length** | **Average Step Time w/o Guidance** | **Average Step Time w/ Guidance** | **Difference** |
> | --- | --- | --- | --- |
> | 96 | 9.618 | 9.667 | 0.51% |
> | 192 | 9.804 | 9.859 | 0.57% |
> | 336 | 9.893 | 9.979 | 0.86% |
>
> To further reduce latency, we propose two potential strategies:
> (i) Adopte ANN-based retrieval, which reduces retrieval time with negligible accuracy change;
> (ii) Scheduled guidance, applying guidance only in some portion of steps.
>
> **Q2. Robustness analysis under different guidance strengths ($\lambda$) against novel or low-density patterns**
> We conducted additional robustness experiments of guidance scale $\lambda$ against different density level, where density is approximated by the average cosine similarity between test query and the top 10 matched indexes. A lower cosine similarity between query and best matched dataset sample indicates a lower density. The experiment is conducted on ETTh1 dataset with context length 168 and prediction length 192. We classify the lowest 25% from the test dataset as low-density, the highest 25% as high-density, and all the remaining as mid-density.
>
> The MAE results are shown in the table below. Varying $\lambda$ shows clear and same performance pattern across different density level, with optimal values around 0.01-0.02. The performance improvement brought by retrieval guidance on low-density region is the highest, showing both the robustness and advantage of applying retrieval guidance in low-density patterns.
>
> | **Density level / Guidance scale** | **0** | **0.005** | **0.01** | **0.02** | **0.03** | **Best Improvement** |
> | --- | --- | --- | --- | --- | --- | --- |
> | **Low** | 0.535  | 0.527  | 0.521  | 0.516  | 0.518  | 3.47% |
> | **Mid** | 0.405  | 0.402  | 0.400  | 0.401  | 0.409  | 1.36% |
> | **High** | 0.415  | 0.411  | 0.408  | 0.408  | 0.414  | 1.72% |
>
> **Q3. Discussions on similarity metrics for retrieval**
>
> Originally, we conducted experiment using cosine similarity in latent space to measure similarity. We have also conducted experiments using two alternative similarity measures: L2 distance and Manhattan distance
>
> These experiments cover both geometric distances and direction-based similarities, and are performed on both the encoded representations and the preprocessed raw sequences. The experiments are conducted on ETTh1 dataset with context length 168 and prediction length 192, using L2 as the energy function for retrieval guidance. We will include the quantitative results as tables and corresponding figures in the appendix. Empirically, our retrieval method is robust to different similarity measures, showing similar performance pattern regarding guidance scale and cosine similarity in latent space consistently provided the most stable and competitive performance, while other measures were slightly inferior or more sensitive to noise. The table reports MAE.
>
> | **Similarity Measure / Guidance Scale** | **0** | **0.005** |  **0.01** |  **0.02** |  **0.03** |
> | --- | --- | --- | --- | --- | --- |
> | **Cosine** | 0.432 | 0.430 | 0.429 | 0.429 | 0.433 |
> | **L2** | 0.432 | 0.430 | 0.429 | 0.431 | 0.436 |
> | **Manhattan** | 0.432 | 0.430 | 0.429 | 0.431 | 0.435 |

---

> ### Author Response · Authors · 2025-12-03
>
> **Q4. A feasible strategy for dynamically updating the retrieval database**
>
> We agree that practical long-term deployment requires an efficient mechanism to update and manage the retrieval database under non-stationary data. Importantly, MiDDiR does not require joint retraining when the database is updated: once the encoder is trained, new observations can be encoded and inserted independently.
>
> A feasible strategy is as follows. In an online setting, each newly observed history–future pair is passed through the frozen encoder once and its latent representation, together with the corresponding target segment, is appended to the retrieval store. To keep the database bounded and adaptive, one can maintain (i) a recency-aware buffer that prioritizes recent samples (e.g., via a sliding time window or time-decay weighting), and (ii) a diversity mechanism (e.g., clustering or density-based pruning) that preserves representative samples from older regimes, including rare but recurrent patterns. The retrieval index (either exact or approximate) can then be updated periodically in the background using the current buffer, without interrupting inference.
>
> This scheme ensures that the retrieval database gradually tracks the evolving data distribution while remaining of controlled size. It is compatible with both exact search and standard approximate nearest-neighbor libraries, and it allows MiDDiR to be deployed in streaming or non-stationary environments without incurring re-computation or memory costs.
>
> ---
>
> We sincerely thank the reviewer again for the thoughtful comments and constructive suggestions. We hope these clarifications have resolve all concerns.

---

### Official Review · Reviewer_ozaX · 2025-11-11

**Soundness:** 2
**Presentation:** 3
**Contribution:** 3
**Rating:** 6
**Confidence:** 3

**Summary:**

The paper proposes a diffusion model-based generative model to forecast time series data, with two key features: (1) a channel dependent encoder which models the dependencies between the input channels, and (2) retrieval guidance, to guide the model predictions to capture the low density regions of the forecasting dataset. The authors compare their proposed method on widely-used forecasting datasets, against a few time series forecasting baselines.

**Strengths:**

To the best of my knowledge, the proposed method is novel, and on the compared datasets, against the compared baselines, the proposed methods seems to perform well.

**Weaknesses:**

I believe this is a good paper, however, I have a few recommendations, which I believe would significantly improve the paper in its current forms:

1. **More baselines**. I believe comparing the proposed model against state-of-the-art time series foundation models (e.g., Chronos-1/2, TimesFM, Tirex, etc.), some of which provide distributional forecasts, will make the results more convincing.

2. **Better datasets**. The datasets used (e.g., ETT, Exchange, Weather) are known to have limited diversity and issues. A few new benchmarks such as GIFT-Eval and fev-bench were released to partially address some of these gaps. I would highly recommend that the authors supplement their results with findings from these benchmarks. Moreover, one of the key features of the proposed study is modeling inter-channel dependencies. However, prior work such as [1] and Chronos-2 have also reported that multivariate modeling yields little or no benefit on these public datasets, which weakens the empirical claims around MiDDiR's improvements.

###
1. Żukowska, Nina, et al. "Towards long-context time series foundation models." arXiv preprint arXiv:2409.13530 (2024).
Goswami, Mononito, et al. "Moment: A family of open time-series foundation models." arXiv preprint arXiv:2402.03885 (2024).

**Questions:**

I do not have any specific questions for the authors.

---

> ### Author Response · Authors · 2025-12-03
>
> We sincerely thank the reviewer for the thoughtful and constructive feedback. We appreciate the clear articulation of strengths and the helpful suggestions for additional analyses and comparisons. Below we address the reviewer’s concerns in detail.
>
> **W1&W2. More baselines & datasets**
>
> We appreciate the reviewer’s suggestion on the baselines and the recent benchmarks. We have conducted additional experiments on GIFT-Eval, a diverse and challenging evaluation suite accompanied with multiple well-established baselines including those suggested by the reviewer. We filtered and adopted the multi-variate subset in GIFT-Eval, to align with our experimental problem setting. The corresponding results have been added in Appendix, and MiDDiR demonstrates competitive results over strong baselines, ranking the 1st on MAPE metric among all non-fundamental deep learning models on medium and long horizon, and ranking the second on short horizon, confirming that the method generalizes well beyond the previous datasets. When compared with all time series models (including fundamental models), MiDDiR ranked the 3rd on MSE out of all 39 models in comparison, surpassing Chronos-2, TiRex and TimesFM. On CRPS metrics, MiDDiR ranked the 10th out of all 39 baselines, which is competitive considering the much lower computational resource consumed for training MiDDiR than those fundamental time series models.
>
> With respect to the reviewer’s concern about limited benefits from multivariate modeling, we would like to clarify an important distinction. Prior work—including Chronos-2 and the analysis in Żukowska et al. (2024)—primarily examines multivariate encoder architectures or joint context representations and often concludes that naive multivariate modeling provides marginal gains on public datasets. Our findings do not conflict with these results. In fact, our own ablations show that simple cross-channel modeling yields limited improvements.
>
> However, MiDDiR leverages inter-channel dependencies in a different and more targeted way:
>
> - Mixed channel-dependency diffusion reduces the difficulty of estimating the joint distribution by decomposing channel-wise and cross-channel interactions.
> - Retrieval guidance operates at the per-channel level, allowing each variable to selectively leverage past patterns that are most relevant to it.
>
> As visualized in Fig. 4 and supported by our ablations, certain variables in both ETTh1 and Weather exhibit strong and meaningful cross-channel dependencies (e.g., Channel 1 ↔ 3 and 2/4 → 5 in ETTh1), while some others behave nearly independently. MiDDiR is designed to respect this heterogeneity rather than assuming full multivariate coupling or full independence.
>
> Thus, our results suggest not that “multivariate modeling always helps,” but rather that properly structured and selective modeling of inter-channel dependencies is beneficial, even on datasets where naïve multivariate architectures show little improvement.
>
> ---
> We sincerely thank the reviewer again for the thoughtful comments and constructive suggestions. Hope these clarifications have resolve all concerns.

---

### Official Review · Reviewer_YiMH · 2025-11-11

**Soundness:** 2
**Presentation:** 3
**Contribution:** 2
**Rating:** 2
**Confidence:** 3

**Summary:**

This paper proposes a novel mixed channel dependency method on time series diffusion model, encoding historical time series in a channel-dependent manner to obtain informative historical representation while denoising in a channel-independent manner to decrease modeling complexity. During inference, we retrieve similar history occurrence for explicitly tilting the score estimation as retrieval guidance to enhance forecasting quality.

**Strengths:**

The writing logic is clear, and each major part of the module is clearly described.

The idea of using retrieved samples to improve forecasting is also making sense to the time series forecasting tasks.

**Weaknesses:**

The simulations are not persuasive enough. The Fig. 4's attention map does not reveal many insights of the forecasting task (and there is no units of the color map?).

There are not enough illustrated examples how the retrieved samples look like, and how they would help the forecasting task.

The motivation behind using diffusion model for time series forecasting is not well justified.

Why the Equation (11) is applied to evaluate the similarity score? Has the authors considered other similarity measure?

The complexity of sample retrieval is not reported.

**Questions:**

Please see weakness above.

---

> ### Author Response · Authors · 2025-12-03
>
> We sincerely thank the reviewer for the detailed and constructive feedback. Below, we address each question and clarify the motivations behind our design choices.
>
> **W1. Insights from Fig.4’s attention map for the forecasting task**
>
> We appreciate the reviewer’s feedback. Fig. 4 is intended to qualitatively illustrate how channel dependencies are captured by our encoder. Rather than showing a uniform diagonal-dominant structure, the attention maps reveal heterogeneous dependency patterns across variables.
>
> Specifically, some variables exhibit near-independence from the others—for example, the last channel in ETTh1 and the 15th channel in Weather, both of which show attention concentrated almost entirely on themselves. In contrast, other variables demonstrate strong cross-channel dependencies: in ETTh1, the 1st and 3rd channels attend strongly to each other, while the 2nd and 4th channels both rely heavily on information from the 5th channel. These non-trivial patterns reflect meaningful inter-variable relationships and match the intuition that certain variables act as key drivers in multivariate dynamics.
>
> To avoid ambiguity, we have added clarification of the color bar and expanded the accompanying explanation to clearly highlight these varying dependency structures and how they support the design of our channel-dependent encoder.
>
> **W2. Extended illustration of retrieved samples and how they help forecasting task**
>
> We thank the reviewer for the suggestion. In Fig. 3, the visualization already demonstrates how retrieval guidance affects generation: the blue dashed line represents the retrieved target sequence, the orange line is the ground truth, and the solid blue line denotes our model's forecast. As the guidance strength increases, the forecast moves closer to the retrieved pattern while preserving consistency with the ground truth.
>
> To further clarify the effect of retrieval, we will include additional examples in the appendix showing:
> (i) the top-$K$ retrieved samples for different datasets and channels, and
> (ii) their similarity scores relative to the query.
> These visualizations will make clear how the retrieved patterns provide useful priors that steer the generation toward historically plausible trajectories.
>
> **W3. Motivation behind using diffusion model for time series forecasting**
>
> Diffusion model has been proven as a strong probabilistic distribution estimator. Time series forecasting can be considered as fitting a conditional probabilistic distribution. Diffusion models have been widely adopted for time series forecasting and demonstrated outstanding results.
>
> Our choice of diffusion models is motivated by both the problem formulation and prior empirical evidence. Forecasting in our setting is explicitly framed as learning the conditional distribution $q(X_p|X_o)$ over high-dimensional, long-horizon multivariate sequences (Section 3.1). Diffusion models are strong probabilistic density estimators for complex, high-dimensional distributions and have shown state-of-the-art performance in conditional generation tasks beyond vision.
>
> In time series specifically, several recent works have successfully adapted diffusion/score-based models to probabilistic forecasting and imputation, achieving competitive or superior performance compared to normalizing flows and autoregressive likelihood models (e.g., TimeDiff, NsDiff). Our method builds on this line of work, using a mixed channel-dependency diffusion architecture to reduce distributional modeling complexity and introducing retrieval guidance to explicitly tilt the diffusion score toward rare but repeated patterns.

---

> ### Author Response · Authors · 2025-12-03
>
> **W4-1. Rational behind Equation (11) for similarity score evaluation**
>
> In Eq. (11), we use cosine similarity between the latent representations of historical sequences to evaluate their similarity. The encoder is trained to organize sequences with similar temporal patterns into similar latent representations. Cosine similarity naturally measures this alignment, making it well suited for retrieving sequences that share structural dynamics (e.g., trend shape or oscillatory pattern) rather than purely matching absolute values.
>
> **W4-2. Other similarity measures**
>
> Originally, we conducted experiment using cosine similarity in latent space to measure similarity. We have also conducted experiments using two alternative similarity measures: L2 distance and Manhattan distance
>
> These experiments cover both geometric distances and direction-based similarities, and are performed on both the encoded representations and the preprocessed raw sequences. The experiments are conducted on ETTh1 dataset with context length 168 and prediction length 192. We will report the quantitative results in MAE as tables and corresponding figures in the appendix. Empirically, our retrieval method is robust to different similarity measures, showing similar performance pattern regarding guidance scale, and cosine similarity provided the most competitive performance, while other measures were slightly inferior.
>
> | **Similarity Measure / Guidance Scale** | **0** | **0.005** |  **0.01** |  **0.02** |  **0.03** |
> | --- | --- | --- | --- | --- | --- |
> | **Cosine** | 0.432 | 0.430 | 0.429 | 0.429 | 0.433 |
> | **L2** | 0.432 | 0.430 | 0.429 | 0.431 | 0.436 |
> | **Manhattan** | 0.432 | 0.430 | 0.429 | 0.431 | 0.435 |
>
> **W5. Analysis on the complexity of sample retrieval**
>
> We have additionally measured the actual runtime of the retrieval step across datasets of different sizes. The results show that retrieval introduces only a small overhead compared with forecasting, and the increase in runtime scales with the size of the retrieval database. We have included these results in the appendix.
>
> | **Dataset** | **ETTh1/ETTh2** | **ETTm1/ETTm2** | **Weather** | **Electricity** | **Traffic** |
> | --- | --- | --- | --- | --- | --- |
> | **# Train Samples** | 8377 | 34297 | 36624 | 18149 | 12017 |
> | **# Test Samples** | 2785 | 11425 | 10444 | 5165 | 3413 |
> | **# Variables** | 7 | 7 | 21 | 370 | 862 |
> | **Retrieval Index Size** | 58464 | 239904 | 768768 | 5824224 | 10344000 |
> | **Average Search Time Per Sample (ms)** | 0.368 | 0.498 | 2.142 | 37.047 | 151.864 |
> | **Average Search Time Per Variable (ms)** | 0.053 | 0.071 | 0.102 | 0.115 | 0.176 |
>
> ---
>
> We sincerely thank the reviewer again for the thoughtful comments and constructive suggestions. Hope these clarifications have resolve all concerns.

---

### Meta-Review · Area_Chair_RqFw · 2026-01-05

**Summary:**

While the paper addresses an interesting and timely problem by combining diffusion-based forecasting with cross-channel dependency modeling and retrieval guidance, several concerns remain regarding its technical motivation and empirical validation. In particular, the use of diffusion models is insufficiently justified over simpler alternatives, the claimed ability to capture inter-channel dependencies is not convincingly supported by results on high-dimensional datasets, and analyses of memory and efficiency are missing. Overall, although the core idea is novel, the paper would benefit from substantially stronger motivation and deeper analysis to clarify when and why the proposed design choices are necessary and effective.

**Reviewer Concerns:**

Reviews points out the insufficient justification and limited experimental results. Authors address the issues by providing extensive experimental evaluations. Detailed explanations also address the issues of clarify. However, the mixed results of extended experimental results would give issue to some reviews.

**Reviewer Scores:**

YiMH and HHK8  would raise the scores with additional experimental evaluations. However, the margin would be limited.

---

### Decision · Program_Chairs · 2026-01-26

Reject